# Federated Learning of Large Language Models with Parameter-Efficient Prompt Tuning and Adaptive Optimization

**Tianshi Che[1][†], Ji Liu[2][†][∗], Yang Zhou[1][∗], Jiaxiang Ren[1],**
**Jiwen Zhou[3], Victor S. Sheng[4], Huaiyu Dai[5], Dejing Dou[6]**

[1] Auburn University, Auburn, United States,

[2] Hithink RoyalFlush Information Network Co., Ltd., Hangzhou, Zhejiang, China,

[3] Baidu Inc. Beijing, China, [4] Texas Tech University, Lubbock, United States,

[5] North Carolina State University, United States, [6] Boston Consulting Group, United States.

## Abstract

Federated learning (FL) is a promising paradigm to enable collaborative model training with decentralized data. However, the training process of Large Language Models (LLMs) generally incurs the update of significant parameters, which limits the applicability of FL techniques to tackle the LLMs in real scenarios. Prompt tuning can significantly reduce the number of parameters to update, but it either incurs performance degradation or low training efficiency. The straightforward utilization of prompt tuning in the FL often raises non-trivial communication costs and dramatically degrades performance. In addition, the decentralized data is generally non-Independent and Identically Distributed (non-IID), which brings client drift problems and thus poor performance. This paper proposes a Parameter-efficient prompt Tuning approach with Adaptive Optimization, i.e., FedPepTAO, to enable efficient and effective FL of LLMs. First, an efficient partial prompt tuning approach is proposed to improve performance and efficiency simultaneously. Second, a novel adaptive optimization method is developed to address the client drift problems on both the device and server sides to enhance performance further. Extensive experiments based on 10 datasets demonstrate the superb performance (up to 60.8% in terms of accuracy) and efficiency (up to 97.59% in terms of training time) of FedPepTAO compared with 9 baseline approaches. Our code is available at https://github.com/llm-eff/FedPepTAO.

## 1 Introduction

As a promising paradigm to handle decentralized data, Federated Learning (FL) (Kairouz et al., 2021) enables collaborative model training without transferring the raw data across a massive number of devices. As a bunch of legal restrictions

---

[†] Equal contribution.
[∗] Corresponding author: jiliuwork@gmail.com, yangzhou@auburn.edu

(Official Journal of the European Union, 2016; Californians for Consumer Privacy, 2020) have been implemented, aggregating the decentralized data into a central server or data center becomes complicated or even impossible (Yang et al., 2019). FL generally exploits a parameter server module (Liu et al., 2023b) to manage the distributed model updates in devices, which only exchanges the parameters of the updated models instead of the raw data, between the parameter server and devices.

Large Language Models (LLMs) (Devlin et al., 2018; Liu et al., 2019b; Brown et al., 2020; Lewis et al., 2019) have achieved major advances in Natural Language Processing (NLP) tasks. The scale of LLMs can range from 110 million parameters to 175 billion parameters, which correspond to huge communication and computation costs to update parameters during the pre-training process (Sanh et al., 2022; Wang et al., 2022) or fine-tuning process (Ding et al., 2023). Both pre-training and fine-tuning update the whole set of parameters of the language model. Thus, the application of FL in the pre-training or the fine-tuning process is almost impossible due to significant communication burden brought by large amount of parameters.

Prompt design (Brown et al., 2020) can lead to excellent performance while freezing the original LLMs. When the LLMs are frozen, only the prompts or prefix are updated during the tuning process, which can significantly reduce the number of parameters to update. For instance, for a sample from a sentiment analysis task (e.g., "beautiful place!"), a discrete prompt "It was [MASK]." for prompt tuning (Brown et al., 2020) and continuous task-specific vectors for prefix tuning (Li and Liang, 2021) can be concatenated to be sent to a LLM, which generates the label of the sample to be "terrible" or "great".

Numerous parameter-efficient prompt or prefix tuning approaches have been proposed to tune the large language models through updating a few train-

able parameters while achieving comparable performance compared with fine-tuning. In order to avoid human involvement in the prompt design, prompt tuning methods (Shin et al., 2020) are proposed to search proper prompts within a discrete space of words, which corresponds to inferior performance compared with fine-tuning. Continuous prompts, i.e., prefix tuning, can be updated to achieve better performance (Liu et al., 2021; Lester et al., 2021). However, this approach leads to sub-optimal performance for the models with less than 10 billion parameters. Although P-tuning V2 (Liu et al., 2022d) achieves comparable performance compared with fine-tuning, it introduces more parameters, which may correspond to heavier communication costs in the setting of FL compared with other parameter-efficient tuning approaches. Some other parameter-efficient prompt tuning methods either suffer from low performance with the focus on low-rank hyper-complex adapter layers (Karimi Mahabadi et al., 2021a) or prompt with a single layer (Liu et al., 2022c), or introduce extra computation costs with attentions (Asai et al., 2022). In addition, existing FL techniques for fine-tuning large language models typically incur performance degradation or low efficiency due to huge communication costs (Tian et al., 2022; Sun et al., 2022; Zhao et al., 2023).

Adaptive optimization methods, e.g., Adaptive Moment Estimation (Adam) and Stochastic Gradient Descent (SGD) with Momentum (SGDM) (Sutskever et al., 2013), have been utilized either on server side (Duchi et al., 2011; Reddi et al., 2018a) or on device side (Yuan et al., 2021; Liu et al., 2020; Gao et al., 2021a; Wang et al., 2020) to achieve superior performance in FL. However, the direct application of the adaptive optimization methods may incur problems of the convergence within the training process (Reddi et al., 2018b). Furthermore, the application of adaptive optimization on a single side, i.e., either device or server, may correspond to poor performance. However, when the adaptive optimization is exploited on both sides (Jin et al., 2022a) may incur heavy communication costs. In addition, client drift (Karimireddy et al., 2020b) may exist in terms of the adaptive optimization due to non-Independent and Identically Distributed (non-IID) data among devices.

In this paper, we propose a Parameter-efficient prompt Tuning approach with Adaptive Optimization, i.e., FedPepTAO, to tune large language models with FL. As transferring the whole set of parameters in all the prompt layers corresponds to heavy communication costs, we propose an efficient and effective method to choose proper layers of prompts based on the importance of each layer. We design a scoring method to identify the importance of each layer according to the tuning impact of the layer on the final convergence accuracy. In addition, we propose an adaptive optimization method on both server side and device side with control measures on each device to achieve superb accuracy. We summarize out major contributions as follows:

- We propose a novel parameter-efficient prompt tuning method with an efficient and effective method to choose proper layers of prompts for FL. The subset of layers of prompts can reduce both the communication and computation costs within FL.

- We provide an original adaptive optimization method on both server side and device side with control measures on each device.

- We carry out extensive experimentation based on 10 datasets, which demonstrates the advantages of FedPepTAO in terms of accuracy (up to 60.8% higher) and efficiency (up to 97.59% faster) compared with 9 baseline approaches.

## 2 Related Work

As updating all the parameters of a pre-trained LLM consumes a large amount of memory and computation resources, prompt tuning (Brown et al., 2020) or prefix tuning (Li and Liang, 2021) is proposed to update a few parameters with a frozen language model while achieving comparable performance compared with fine-tuning. While prompt tuning may correspond to inferior performance with discrete space of words (Shin et al., 2020), prefix tuning (Liu et al., 2021; Lester et al., 2021) can deal with continuous prompts to achieve better performance. Adapter modules (Houlsby et al., 2019) are exploited to tune large language models with prompts, which may incur heavy computation costs due to the calculation of feed-forward project and non-linearity or attention mechanism (Asai et al., 2022). Although efficient low-rank hypercomplex mechanism (Karimi Mahabadi et al., 2021b) can be utilized to reduce parameters to update, the performance may degrade. P-tuning V2 achieves comparable performance compared with

fine-tuning (Liu et al., 2022d), which the prompts added into each layer of the large language model. Prompts can be added at a single layer to further reduce the computation costs (Liu et al., 2022c), which may incur performance degradation and depends on multiple trials with each layer. In addition, the selection of the layer may incur long execution time to verify the impact on the final accuracy. Although the NASWOT algorithm (Mellor et al., 2021) can be exploited to analyze the performance of a neural network architecture, it is only compatible with the neural networks based on the ReLU activation function (Nair and Hinton, 2010; Glorot et al., 2011).

Parallel, distributed, and federated learning have been extensively studied in recent years (Liu et al., 2023a; Chen et al., 2023b,a; Lee et al., 2019; Wu et al., 2021; Goswami et al., 2020; Zhang et al., 2021; Zhou et al., 2022; Guo et al., 2022; Jin et al., 2022a; Che et al., 2022; Yan et al., 2022a; Liu et al., 2022b; Yan et al., 2022b,c; Jin et al., 2022b, 2021; Zhao et al., 2021; Zhou and Liu, 2013; Lee et al., 2013; Zhang et al., 2013; Zhou et al., 2014; Zhang et al., 2014; Bao et al., 2015; Zhou et al., 2015a,b; Lee et al., 2015; Jiang et al., 2019; Zhang et al., 2022; Zhou, 2017; Hong et al., 2023; Chen et al., 2018b,a; Gan et al., 2023; Che et al., 2023; Liu et al., 2023c, 2022a; Li et al., 2023; Oliveira et al., 2019; Liu et al., 2019a, 2016, 2015). Some existing FL techniques have been proposed to fine-tuning large language models, which may suffer from performance degradation or low efficiency due to huge communication costs (Tian et al., 2022; Sun et al., 2022; Zhao et al., 2023). FedBert (Tian et al., 2022) exploits split learning to split a model into two parts, i.e., one with transformer and the other one with head and embedding. As the transformer is shared on the server, FedBert may correspond to inferior performance and huge communication costs compared with prompt-tuning or prefix-tuning. Some other FL methods only fine-tune a part of the model weights (Sun et al., 2022), which still suffer from heavy communication costs for big language models. FedPrompt (Zhao et al., 2023) enable FL based on prompt tuning, which communicates the whole set of parameters in prompts corresponding to huge communication costs.

Adaptive Moment Estimation (Adam) and Stochastic Gradient Descent (SGD) with Momentum (SGDM) (Sutskever et al., 2013) are exploited within FL on server side (Duchi et al., 2011; Reddi et al., 2018a) or on device side (Yuan et al., 2021; Liu et al., 2020; Gao et al., 2021a; Wang et al., 2020) to address the client drift problem brought by the non-IID data in FL (Karimireddy et al., 2020b). However, the direct application of adaptive optimization on devices may lead to convergence problem (Reddi et al., 2018b). In addition, the application of adaptive optimization on both server and device sides may incur heavy communication costs (Jin et al., 2022a).

Different from the previous work, we propose a general scoring method to analyze the correlation of each layer and the output of the large language model, which can represent the importance of each layer. Then, we select the prompt parameters of proper layers to be updated with FL while leaving other prompt parameters of other layers to be adjusted locally with a lossless method so as to achieve superb performance with limited communication costs. In addition, we introduce control measures on each device to alleviate the client drift problem and propose a novel adaptive optimization method on both server and device sides to further improve the performance.

## 3    Problem Formulation

The problem to address in this paper is how to efficiently tune a large language model based on prompt tuning in FL. Given a large language model $\mathcal{M}$ with $L$ layers, we add prompts for each layer in $\mathcal{M}$ and denote the set of parameters to generate prompts by $P_l$ with $l$ representing the number of layer in $\mathcal{M}$. The whole set of prompts is denoted by $\mathcal{P}$ During the tuning process, the parameters in $\mathcal{M}$ are frozen and cannot be updated while the parameters in $\mathcal{P}$ are updated to improve the performance of $\mathcal{M}$.

We consider a FL setting with a parameter server and $M$ devices. We assume that the data for the tuning process of $\mathcal{M}$ is distributed among multiple devices. On each Device $i$, a dataset $D_i = \{s_i, m_i\}^{n_i}$ is located with $s_i$, $m_i$, and $n_i$ representing a sample, the corresponding label of $s_i$, and the number of samples in $D_i$. We denote the total number of the samples on all the devices by $N$, the set of all the samples by $S$ and that of labels by $M$. Due to the limited computation capacity, each Device $i$ can only perform the inference of $\mathcal{M}$ while updating $\mathcal{P}_i$ with $\mathcal{P}_i$ representing the prompt parameters in Device $i$. In order to reduce communication costs, we

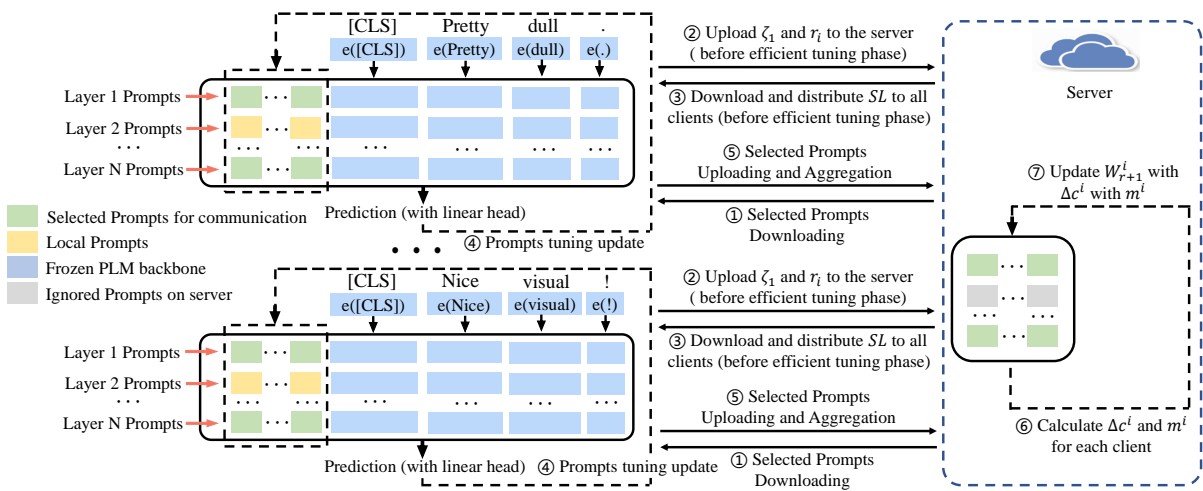

Figure 1: The system model of FedPepTAO.

enable the exchange the parameters of the prompts within a subset of selected layers $SL_i$ between Device $i$ and the parameter server while the other prompt parameters are only updated within each device. We denote the set of prompt parameters in all the devices by $\mathbf{P}$. The problem to address in this paper can be formulated as how to efficiently generate $\mathbf{P}$ such that the global loss is minimized:

$$\min_{\mathbf{P}} \left[ \mathcal{F}(\mathcal{M}, \mathbf{P}) \triangleq \frac{1}{N} \sum_{i=1,\ p_i \in \mathbf{P}}^{M} n_i F_i(\mathcal{M}, p_i) \right],$$ (1)

where $\mathcal{F}(\mathcal{M}, \mathbf{P})$ represents the global loss, $F_i(\mathcal{M}, p_i) \triangleq \frac{1}{n_i} \sum_{\{s_i, m_i\} \in \mathcal{D}_i} f(\mathcal{M}, p_i, s_i, m_i)$ refers to the loss function on Device $k$ with $f(\mathcal{M}, p_i, s_i, m_i)$ calculating the local loss of the combination of the large language model $\mathcal{M}$ and prompt parameters $p_i$ on $\{s_k, m_k\}$.

For NLP tasks, each sample $s_k \in S$ is the input of the large language model and $m_k \in M$ is the corresponding label. Each sample $s_k$ is composed of multiple tokens, i.e., $s_k = \{s_k^1, s_k^2, ..., s_k^t\}$, where $t$ represents the length of the input. The prompt $p$ consists of multiple tokens $p = \{p_1, p_2, ..., p_h\}$, and the corresponding prompt parameters can be trained. The prompts differ according to layers. We denote the template by $\mathcal{T}(\cdot)$, which defines how to concatenate the input tokens with the prompt. For instance, $s_k^p = \mathcal{T}(s_k, p)$ represents the sample combined with the prompt, which contains one [MASK] token. The output of the large language model with the prompts predicts the label $m_k$, which corresponds to the [MASK] token after applying a verbalizer $\mathcal{V}(\cdot)$, i.e., $\hat{m}_k = \mathcal{V}(o_k)$ with $o_k$ representing the output of the model and $\hat{m}_k$ referring to the predicted label.

In this section, we first present the system model

of FedPepTAO. Then, we propose parameter-efficient prompt tuning method and the adaptive optimization method, respectively.

### 3.1 System Model

As shown in Figure 1, we consider a parameter server and multiple devices for the tuning process of FedPepTAO. We assume that a large language model is deployed on each device. For each layer, we insert a prompt module. During the tuning process, the large language model only perform inference while the prompt modules of each layer perform both the inference of the input and the update of parameters. Within the FL tuning process, the prompt parameters of specific layers are communicated between the device and the server.

During the FL tuning process of FedPepTAO, the prompt parameters in each device are updated with multiple rounds. Each round consists of five steps. First, a set of devices are selected to perform the update of prompt parameters. Second, these devices receive the corresponding updated prompt parameters of specific layers from the server (①). The selection of the specific layers is based on our parameter-efficient prompt tuning method (see details in Section 3.2) (② - ③). Third, the prompt parameters are updated with our adaptive optimization method (see details in Section 3.3) based on the data on each device (④). Fourth, the prompt parameters of specific layers are sent back to the server (⑤). Fifth, the prompt parameters are aggregated on the server with the adaptive optimization method (⑥ - ⑦).

### 3.2 Parameter-efficient Prompt Tuning

We propose a parameter-efficient prompt tuning method to efficiently tune the language model with

FL. Instead of synchronizing the full set of prompt parameters, we select a proper set of layers for each device and only exchange the prompt parameters of these layers during the tuning process. In this section, we propose a scoring method to measure the importance of each layer. Then, we propose a lossless layer selection method to select the proper layers, which reduces the communication costs without performance degradation.

Given the prompt parameters based on any activation function, we can calculate the hidden states of each parameter at each layer of the large language model. With a batch of local data samples $S_i = \{s_i\}^{n_i}$ mapped through the large language model and the prompt parameters corresponding to the function $f_p(s_i)$, the hidden state corresponding to Node $k$ at $l$-th layer is $f_{p_{k,l}}(s_i)$. Then, the hidden states of Layer $l$ corresponding to Sample $s_i$ is $\mathbf{h}_{i,l} = \{f_{p_{1,l}}(s_i), f_{p_{2,l}}(s_i), ..., f_{p_{K_l,l}}(s_i)\}$, with $K_l$ representing the number of nodes at Layer $L$.

As the difficulty for a network to learn to separate the input samples has positive correlation with the similarity of the hidden states (Mellor et al., 2021), we examine the correlation between the hidden states of any two layers by computing the following kernel matrix:

$$\mathbf{K}_{h_i} = \begin{pmatrix} Cos(\mathbf{h}_{i,1}, \mathbf{h}_{i,1}) & \cdots & Cos(\mathbf{h}_{i,1}, \mathbf{h}_{i,L}) \\ \vdots & \ddots & \vdots \\ Cos(\mathbf{h}_{i,L}, \mathbf{h}_{i,1}) & \cdots & Cos(\mathbf{h}_{i,L}, \mathbf{h}_{i,L}) \end{pmatrix} \quad (2)$$

where $Cos(\mathbf{h}_{i,L}, \mathbf{h}_{i,1})$ represents the cosine similarity between two vectors (Dehak et al., 2010). Then, we calculate the eigenvalues of the kernel matrix $\Lambda_i = \{\lambda_{i,1}, \lambda_{i,2}, .., \lambda_{i,L}\}$, with $\lambda_l$ representing the distinction of Layer $l$ compared with other layers based on Sample $s_i$. Afterward, we compute the score ($\zeta_{i,l}$) of Layer $l$ with the local dataset on Device $i$ using the Formula 3, which can avoid the possibility of unacceptable performance penalty due to abnormal eigenvalues (Gao et al., 2021b).

$$\zeta_{i,l} = \frac{1}{n_i} \sum_{j=1}^{n_i} log(\lambda_{j,l} + \epsilon) + (\lambda_{j,l} + \epsilon)^{-1}, \quad (3)$$

where $\epsilon$ refers to a small positive value, e.g., $1 * e^{-5}$. We calculate the global score of each layer leveraging Formula 4 with $\gamma_i = \zeta_{i,l}$.

$$\gamma = \sum_{i=1}^{N} \frac{n_i}{N} \gamma_i, \quad (4)$$

**Algorithm 1** Federated Parameter-efficient Prompt Tuning

**Require:**
    $L$: The list of layers in a large language model
    $M$: The set of devices
    $w$: The prompt parameters of the initial model
    $w^t$: The prompt parameters of the current model in Round $t$

**Ensure:**
    $SL$: The set of selected layers
1:   $SL \leftarrow \emptyset$
2:   **for** $i \in M$ (on each device) **do**
3:       $\Delta_i \leftarrow w - w^t$
4:       $H(w_i^t) \leftarrow$ Get Hessian matrix of $w_i^t$
5:       $\Lambda^{H(w_i^t)} \leftarrow$ Eigenvalues($H(w_i^t)$)
6:       $\{\lambda_1^H, \lambda_2^H, ..., \lambda_{K_i}^H\} \leftarrow$ Sort in ascending order of $\Lambda^{H(w_i^t)}$
7:       $\mathcal{B}(\Delta_i) \leftarrow H(w_i^t) - \nabla F_i(\Delta_i + w_i^t)$
8:       $\mathscr{L}_i \leftarrow$ Get Lipschitz constant of $\mathcal{B}(\Delta_i)$
9:       $k_i \leftarrow$ Get the first $k$ that meets $\lambda_{k+1}^H - \lambda_k^H > 4\mathscr{L}_i$
10:     $r_i \leftarrow \frac{K_i - k_i}{K_i}$
11:     **for** $l \in L$ **do**
12:         Calculate $\zeta_{i,l}$ according to Formula 3
13:     **end for**
14: **end for**
15: Aggregate $r$ and each $\zeta_l$ based on Formula 4
16: $\zeta \leftarrow$ Sort $\{\zeta_1, \zeta_2, ..., \zeta_L\}$ in descending order
17: **while** $l \in \zeta$ and $\frac{Para(SL)}{Para(L)} < r$ **do**
18:     $SL \leftarrow SL \cup l$
19: **end while**

where $\gamma$ represents the variable.

In order to efficiently tune the large language model without performance degradation, we exploit a lossless method as shown in Algorithm 1 to select the set of proper layers within FL. Within first $t$ rounds, the prompt parameters of all the layers are communicated between the server and each device. $t$ can be small, e.g., 5 or 10. At $t$-th round, we perform the layer selection. First, we calculate $\Delta_i$ as the changement of the prompt parameters (Line 3). Then, we calculate the Hessian matrix (based an efficient PyHessian library (Yao et al., 2020)) of the current model (Line 4), the corresponding eigenvalues (Line 5), and sort the eigenvalues in ascending order (Line 6). Afterward, we construct a base function in Line 7 with $\nabla F_i$ representing the gradients and calculate the Lipschitz constant of the base function 8. We take the first $k$ that can meet the constraint in Line 9, and calculate the minimum remaining prompt parameter ratio in

the selected layers $\mathcal{R}_i$, inspired by (Zhang et al., 2021), which can achieve lossless compared with those at all the layers. We calculate the score of each layer in Lines 11 - 13. The execute of Lines 3 - 13 can be carried out in parallel on each device. We aggregate the prompt parameter ratio and the scores based on Formula 4 from each device to the server (Line 15). Then, we sort the layers according to the scores in descending order (Line 16). Finally, we add the layers into the selected layer set based on the scores in descending order (Lines 17 - 19), with $Para(SL_i)$ representing the number of parameters in the selected layer set. In the following rounds, the prompt parameters in $SL$ are communicated between devices and the server.

### 3.3 Communication-Efficient Adaptive Optimization

While data is generally non-IID, we propose a novel communication-efficient adaptive optimization to achieve superb performance without introducing extra communication costs. In order to achieve excellent performance, we propose applying adaptive optimization on both server based on momentum (Cutkosky and Mehta, 2020) and device sides based on Adam (Kingma and Ba, 2015). We reset the first and the second momentum buffers to zero at the beginning of local update (Wang et al., 2021) to avoid extra communication of the momentum variables between the server and the device. In addition, we maintain a state for each device on the server to avoid possible client drift problem incurred by non-IID (Karimireddy et al., 2020b).

The algorithm of communication-efficient adaptive optimization for FL prompt tuning is shown in Algorithm 2. Within each round, we first randomly sample a subset of devices (Line 3). Then, the prompt parameters corresponding to the model in the last round is sent to each device (Line 5), and each selected device perform local update based on Adam (Lines 7 - 8). Afterward, each selected device returns the accumulated difference of the prompt parameters to the server (Line 10). Please note that the execution of Lines 4 - 11 can be performed in parallel on each selected device. We aggregate the differences based on Formula 4 (Line 12). Inspired by (Karimireddy et al., 2020b), we calculate the control variate $c_i^r$ (Line 14) and the corresponding difference $\Delta c_i^r$ (Line 15) for each device on the server. We aggregate the control variate differences based on Formula 4 (Line 17), and

---

**Algorithm 2** Communication-Efficient Adaptive Optimization

---

**Require:**
  $M$: The set of devices
  $w$: The prompt parameters of the initial model
  $R$: The maximum number of global round
  $\alpha$: The local step size
  $\beta$: The momentum parameter
  $\eta = \{\eta^1, \eta^2, ..., \eta^R\}$: The set of global learning rates
  $T = \{T_1, T_2, ..., T_M\}$: The set of local epoch $T_i$ on each Device $i$

**Ensure:**
  $w^R$: The final model
1: $w^0 \leftarrow w, c_i^0 \leftarrow 0, c_g^0 \leftarrow 0, m_i^0 \leftarrow 0, \forall\, i \in M$
2: **for** $r = 1, \cdots, R$ **do**
3:   Randomly sample a subset $S$ of devices $M$
4:   **for** $i \in S$ (on each device) **do**
5:     $w_i^{r,0} \leftarrow w^{r-1}, m_i^{r,0} \leftarrow 0, v_i^{r,0} \leftarrow 0$
6:     **for** $t = 1, 2, \cdots, T_i$ **do**
7:       $g_i^{r,t} \leftarrow \nabla_{w_i^{r,t-1}} F_i(w_i^{r,t-1})$
8:       $w_i^{r,t}, m_i^{r,t}, v_i^{r,t} \leftarrow$ Adam update with $w_i^{r,t-1}, m_i^{r,t-1}, v_i^{r,t-1}$, and $g_i^{r,t}$
9:     **end for**
10:    $\Delta w_i^r = w_i^{r,T_i} - w_i^{r,0}$
11:   **end for**
12:   Aggregate $\Delta w^r$ based on Formula 4
13:   **for** $i \in S$ (on the server) **do**
14:     $c_i^r = c_i^{r-1} - c_g^{r-1} - \frac{1}{T^i \alpha} * \Delta w_i^r$
15:     $\Delta c_i^r = c_i^r - c_i^{r-1}$
16:   **end for**
17:   Aggregate $\Delta c^r$ based on Formula 4
18:   $c_g^r = c_g^{r-1} + \Delta c^r * \frac{|S|}{|M|}$
19:   $g_g^r = \Delta w^{r-1} - \Delta w^r$
20:   **for** $i \in S$ (on the server) **do**
21:     $m_i^r = \beta * m^{r-1} + (1 - \beta) * g_g^r + c_g^r - c_i^r$
22:     $w_i^r = w^{r-1} - \eta^r * m_i^r$
23:   **end for**
24:   Aggregate $w^r$ based on Formula 4
25:   Aggregate $m^r$ based on Formula 4
26: **end for**

---

calculate the global control variate (Line 18) and global gradients (Line 19). Afterword, we update the momentum (Line 21) and prompt parameters (Line 22) for each selected device. Finally, we aggregate the global prompt parameters (Line 24) and momentum (Line 25). The communication cost of Algorithm 2 depends on the size of prompt parameters, which is similar to that of FedAvg (McMahan et al., 2017), while achieving superb performance.

| Method | Comm. Params | QNLI | SST-2 | CoLA | MPRC | RTE | BoolQ | MPQA | Subj | Trec | MR | Avg |
|---|---|---|---|---|---|---|---|---|---|---|---|---|
| Adapter | 7.4M | 87.79 | 94.04 | 30.96 | 71.81 | 68.59 | 75.11 | 90.97 | 94.6 | 79 | _91.9_ | 75.36 |
| FedPrompt | 131K | 85.91 | 94.84 | 33.05 | 77.87 | 61.73 | 74.77 | 90.45 | 94.25 | 95 | 91.65 | 76.85 |
| P-tuning v2 | 6.3M | 85.19 | _95.3_ | 41.82 | _82.78_ | 79.42 | _79.66_ | 91 | _96.9_ | 96.4 | 91.45 | 82.05 |
| Prompt Tuning | 20K | 51.62 | 61.01 | 3.36 | 48.04 | 52.35 | 59.72 | 81.65 | 65.2 | 36.4 | 63.25 | 42.56 |
| IDPG | 137K | 72.2 | 93.01 | 4.59 | 70.83 | 70.4 | 71.96 | 90.07 | 94 | 78.4 | 91.4 | 60.63 |
| ATTEMPT | 207k | 54.93 | 85.89 | 4.63 | 78.65 | 58.48 | 73.49 | _91.05_ | 88.95 | 82.2 | _91.9_ | 58.28 |
| LPT | 792k | _89.2_ | 94.84 | _53.7_ | 82.07 | 79.7 | 62.7 | 90.55 | 96.5 | 96.4 | 91.4 | 82.35 |
| MomD | 6.3M | 67.42 | 93.92 | 1.64 | 75.17 | 75.45 | 62.02 | 89.05 | 49.95 | 38.6 | 83 | 63.62 |
| MomS+AdamD | 6.3M | 87.85 | 95.18 | 42.96 | 80.15 | _82.31_ | 78.1 | 90.95 | 96.75 | _96.8_ | 91.55 | _84.26_ |
| FedPepTAO | 492K | **89.57** | **95.87** | **56.35** | **87.52** | **85.56** | **79.72** | **91.4** | **97.1** | **97.2** | **93** | **86.4** |

Table 1: The accuracy with FedPepTAO and diverse baseline approaches. All the methods from GLUE benchmark are evaluated on development sets while other tasks are evaluated with test sets. The best results are highlighted in **bold** and the second bests are marked with underline. All the results are obtained using RoBERTa$_{LARGE}$.

# 4 Experiments

In this section, we present the experimental results over 9 baselines and 10 commonly-used tasks to demonstrate the advantages of FedPepTAO.

## 4.1 Experimental Setup

We consider an FL environment with 100 devices and a parameter server. In each epoch, we randomly sample 10 devices to perform the local update. We exploit 10 widely used NLP tasks including QNLI (Rajpurkar et al., 2016), SST-2 (Socher et al., 2013), CoLA (Warstadt et al., 2019), MRPC (Dolan and Brockett, 2005), RTE (Giampiccolo et al., 2007), and BoolQ (Clark et al., 2019) from the GLUE benchmark, and 4 other tasks including MPQA (Wiebe et al., 2005), Subj (Pang and Lee, 2004), TREC (Voorhees and Tice, 2000), and MR (Pang and Lee, 2005) (Please see details in Appendix A.2). We take 9 existing approaches as baselines, including an adapter-based method, i.e., Adapter (Houlsby et al., 2019), 6 prompt-based tuning methods, i.e., FedPrompt (Zhao et al., 2023), P-tuning v2 (Liu et al., 2022d), Prompt Tuning (Lester et al., 2021), IDPG (Wu et al., 2022), ATTEMPT (Asai et al., 2022), LPT (Liu et al., 2022c), and 2 optimization approaches, i.e., momentum on the device side with simple SGD on the server side (MomD) (Karimireddy et al., 2020a) and momentum on the server side with Adam on the device side without control variate (MomS+AdamD). We adapt the centralized methods, i.e., Adapter, P-tuning v2, Prompt Tuning, IDPG (S-IDPG-PHM), ATTEMPT, and LPT, with FedAvg (McMahan et al., 2017) to the FL setting for a fair comparison.

We evaluate FedPepTAO and all other methods on RoBERTa$_{LARGE}$ (Liu et al., 2019b), which consists of 24 layers of transformers followed by a large language model head and 355M pre-trained parameters. To demonstrate the adaptability of FedPepTAO, we carried out extra experiments with three additional decoder-based models, i.e., GPT2$_{LARGE}$ model (Radford et al., 2019) with 774M parameters on MRPC, MR, SST-2 dataset, LLaMA 3B model (Touvron et al., 2023) on RTE, MRPC dataset, and LLaMA 7B model (Touvron et al., 2023) on MRPC dataset. The backbones of these models are frozen for all methods.

## 4.2 Evaluation of FedPepTAO

As shown in Table 1, FedPepTAO significantly outperforms baseline methods in terms of the best accuracy (up to 25.39%, 23.83%, 14.53%, 60.8%, 51.76%, 51.72%, 17.02%, 54.71%, 13.39% compared with Adapter, FedPrompt, P-tuning v2, Ptompt Tuning, IDPG, ATTEMPT, LTP, MomD, and MomS+AdamD, respectively). In addition, the average of the best accuracy (average accuracy) for each task is shown in the last column. The advantage of FedPepTAO is obvious in terms of the average accuracy as well, i.e., 11.04%, 9.55%, 4.35%, 43.84%, 25.77%, 28.12%, 4.05%, 58.6%, 13.39%, higher compared with Adapter, FedPrompt, P-tuning v2, Ptompt Tuning, IDPG, ATTEMPT, LTP, MomD, and MomS+AdamD, respectively. Although FedPrompt, Prompt Tuning, IDPG, and ATTEMPT exploit fewer parameters, the corresponding accuracy is inferior. FedPrompt, Prompt Tuning, and ATTEMPT only update the soft prompt for the first layer, which cannot optimize other important layers and incurs sub-optimal performance. IDPG shares a single generator for each layer, which cannot address the characteristics of diverse layers and leads to inferior accuracy. Different from these methods, FedPepTAO can well optimize the prompt parameters for each layer based on P-tuning v2, while choosing the proper layers for aggregation within FL so as to achieve excellent performance. In addition, we exploit the adaptive optimization on both server and device

| Method | QNLI | SST-2 | CoLA | MPRC | RTE | BoolQ | MPQA | Subj | Trec | MR |
|---|---|---|---|---|---|---|---|---|---|---|
| Adapter | 8096 | 1065 | 2655 | / | 2218 | 937 | 758 | 1178 | 1388 | 1797 |
| FedPrompt | 12987 | 668 | 1471 | 1824 | / | 1485 | 412 | 1284 | 336 | 618 |
| P-tuning v2 | 10780 | **17** | 489 | **154** | 201 | 135 | 105 | 132 | 95 | 748 |
| Prompt Tuning | / | / | / | / | / | / | / | / | / | / |
| IDPG | / | 3322 | / | / | 689 | 3254 | 908 | 1347 | 1220 | 1912 |
| ATTEMPT | / | / | / | 1774 | / | 973 | 438 | 2573 | 1028 | 1221 |
| LPT | 2918 | 650 | 733 | 156 | 270 | / | 162 | 155 | 112 | 860 |
| MomD | / | 1328 | / | / | 838 | / | 537 | / | / | / |
| MomS+AdamD | **697** | 209 | 488 | 1178 | 192 | 139 | 141 | 166 | 100 | **610** |
| FedPepTAO | 781 | 97 | **219** | 230 | **186** | 129 | 31 | 62 | 76 | 610 |

Table 2: The tuning time (s) to achieve a target accuracy (85% for QNLI, 92.5% for SST-2, 3% for CoLA, 77% for MRPC, 65% for RTE, 71% for BoolQ, 85% for MPQA, 88% for Subj, 78% for Trec, 91% for MR) with FedPepTAO and diverse baseline approaches. "/" represents that training does not achieve the target accuracy. The best results are highlighted in **bold** and the second bests are marked with underline. All the results are obtained using RoBERTa$_{LARGE}$.

| Method | MRPC | | MR | | SST-2 | |
|---|---|---|---|---|---|---|
| | Acc | Time | Acc | Time | Acc | Time |
| FedPrompt | 74.98 | / | 57.2 | / | 76.49 | / |
| P-tuning v2 | 74.8 | / | 73 | / | 74.77 | / |
| ATTEMPT | 37.91 | / | 57.3 | / | 85.89 | / |
| LPT | 74.98 | / | 84.8 | 1455 | 77.06 | / |
| MomD | 77.23 | 503 | 85.7 | 1694 | 92.55 | 3638 |
| MomS+AdamD | 76.89 | 291 | 88.2 | 262 | 92.55 | 2488 |
| FedPepTAO | 81.23 | 273 | 89.5 | 222 | 93 | 2248 |

Table 3: Accuracy and tuning time (s) to achieve target accuracy (75% for MRPC, 81% for MR, and 92.5% for SST-2) on GPT2$_{LARGE}$ model. "/" represents that training does not achieve the target accuracy.

| Method | RTE | | MRPC | |
|---|---|---|---|---|
| | Acc | Time | Acc | Time |
| FedPrompt | 78.34 | 540 | 81.86 | 459 |
| P-tuning v2 | 56.68 | / | 75.17 | / |
| ATTEMPT | 64.98 | / | 81.18 | 718 |
| LPT | 64.98 | / | 79.77 | 789 |
| MomD | 80.87 | 360 | 80.26 | 669 |
| MomS+AdamD | 80.87 | 360 | 75.58 | / |
| FedPepTAO | 83.39 | 325 | 86.46 | 409 |

Table 4: Accuracy and tuning time (s) to achieve target accuracy (75% for RTE, 81% for MRPC) on LLaMA 3B model. "/" represents that training does not achieve the target accuracy.

sides to achieve superb accuracy. Compared with Adapter (93.4%), P-tuning v2 (92.19%), and LPT (37.98%), our methods can well reduce the number of parameters to transfer between devices and the server because of the proper layer selection, which corresponds to smaller communication costs. As a result, the efficiency of FedPepTAO is significantly higher than baseline approaches (up to 95.91%, 95.17%, 92.76%, 99%, 97.28%, 97.59%, 85.8%, 94.23%, 80.48%, faster compared with Adapter, FedPrompt, P-tuning v2, Ptompt Tuning, IDPG, ATTEMPT, LTP, MomD, and MomS+AdamD, respectively).

## 4.3 Evaluation of FedPepTAO on Extra LLMs

To demonstrate the adaptability of FedPepTAO, we carried out extra experiments with three additional decoder-based models, i.e., GPT2$_{LARGE}$ model (774M) on MRPC, MR, SST-2 dataset, LLaMA 3B model on RTE, MRPC dataset, and LLaMA 7B model on MRPC dataset.

As shown in Table 3 below, FedPepTAO significantly outperforms baseline methods in terms of the best accuracy on the decoder-based GPT2$_{LARGE}$ model (up to 32.3%, 18.23%, 43.32%, 15.94%, 4%, 4.34% higher compared to FedPrompt, P-tuning v2, ATTEMPT, LPT, MomD and MomS+AdamD,

respectively). Furthermore, the efficiency of FedPepTAO is significantly higher than baseline approaches (up to 84.74%, 86.89%, and 15.27% faster compared to LPT, MomD, and MomS+AdamD, respectively).

When the model becomes larger, i.e., LLaMA 3B with 3 billion parameters, FedPepTAO still achieves the best accuracy (up to 4.6%, 11.29%, 5.28%, 6.69%, 6.2%, 10.88% higher compared to FedPrompt, P-tuning v2, ATTEMPT, LPT, MomD and MomS+AdamD, respectively) and better efficiency (up to 39.81%, 43.04%, 48.16%, 38.86%, 9.72% faster compared to FedPrompt, ATTEMPT, LPT, MomD, and MomS+AdamD, respectively) as illustrated in Table 4.

We verify the performance of our method with another large model, i.e., LLaMA 7B with 7 billion parameters. As shown in Table 5, FedPepTAO outperforms baseline methods in terms of accuracy (up to 5.05%, 26.71%, 18.41%, 18.41%, 6.2%, 10.88% higher compared to FedPrompt, P-tuning v2, ATTEMPT, LPT, MomD and MomS+AdamD, respectively) and efficiency (up to 15.77%, 75.14%, 32.06%, 67.04% faster compared to ATTEMPT, LPT, MomD, and MomS+AdamD, respectively), which demonstrates the scalability of FedPepTAO.

| Method | MRPC | |
|---|---|---|
| | Acc | Time |
| FedPrompt | 76.18 | / |
| P-tuning v2 | 74.98 | / |
| ATTEMPT | 81.52 | 317 |
| LPT | 81.95 | 1074 |
| MomD | 76.2 | 393 |
| MomS+AdamD | 75.75 | 810 |
| FedPepTAO | 82.34 | 267 |

Table 5: Accuracy and tuning time (s) to achieve target accuracy (75% for MRPC) on LLaMA 7B model. "/" represents that training does not achieve the target accuracy.

### 4.4 Evaluation of Parameter-efficient Tuning

Formula 4 calculates the global score $\zeta_l$ of each transformer layer in the model, based on which the proper layers are selected to enable the communication between devices and the server. The selection of layers is critical to the performance of FL. In this section, we compare our parameter-efficient prompt tuning (PEPT) with three other selection strategies, i.e., select with ascending order, descending order ((Liu et al., 2022d)), and random order. Our PEPT method can significantly outperform ascending order (up to 5.78%), descending order (up to 1.81%), and random order (up to 2.89%) (see Figure 5 in Appendix). In addition, we conduct experiments and demonstrate that our PEPT method outperforms random layer selection strategy by 2.93% on RTE, 4.73% on MRPC, and 7.29% on CoLA dataset (see Appendix B.6 for details).

### 4.5 Evaluation of Adaptive Optimization

To demonstrate the effectiveness of Algorithm 2, we compare our adaptive optimization method with six baseline methods, i.e., FedAvg (McMahan et al., 2017), MomD, momentum on the server side with SGD on the device side (MomS) (Reddi et al., 2018a), momentum on the server side with control variate and SGD on the device side (MomS+Con) (Reddi et al., 2018a), Adam on device side with simple SGD on the server side (AdamD), and MomS+AdamD, on the RTE task (see Figure 6 in Appendix). FedPepTAO corresponds to the highest accuracy compared to baseline methods (up to 8.3% compared with MomD, 29.24% compared with MomS, 29.24% compared with MomS+Con, 2.16% compared with AdamD, 5.05% compared with MomS+AdamD, and 29.24% compared with FedAvg). The advantage of FedPepTAO is up to 5.05% compared with MomS+AdamD, which shows that the control variate can avoid client drift

in FL settings and lead to excellent performance. In addition, please note that FedPepTAO calculates the control variate on the server with updated gradients (Algorithm 2), without extra communication cost.

### 4.6 Hyperparameter Evaluation

In this section, we evaluate the performance of FedPepTAO with divers hyperparameters. Additional experiments are in the Appendix B.

**Impact of server learning rate** Due to the high sensitivity of the hyperparameters in NLP tasks, we investigate the impact of divers server learning rates on the RTE dataset. We analyze the accuracy with the learning rates $1e^{-2}, 5e^{-3}, 1e^{-3}, 5e^{-4}$. We find that the best performance was achieved when $lr = 1e^{-3}$, which only slightly outperforms $lr = 5e^{-4}$ by 0.37% (see Figure 2 in the Appendix). This demonstrates that FedPepTAO is easy to fine-tune in practice.

**Impact of heterogeneous data distribution** Data heterogeneity has always been a common challenge in FL. To investigate the robustness of our approach to different degrees of non-IID data, we conduct experiments under various levels of non-IID degrees. We observe that the accuracy of FedPepTAO is the best with $\alpha$ ranging from 1.0 to 5.0 (see Figure 3 in Appendix). A smaller $\alpha$ represents a higher degree of heterogeneity. This demonstrates that our approach is robust to different data heterogeneity.

## 5 Conclusion

In this paper, we propose an original parameter-efficient prompt tuning with adaptive optimization approach for large language models with FL, i.e., FedPepTAO. We dynamically calculate the score of each layer to choose proper layers of prompts for FL without accuracy degradation. In addition, we provide a novel adaptive optimization method with control variate on the server to achieve superb performance without extra communication costs. We carry out extensive experiments based on 10 tasks and 9 state-of-the-art baseline approaches, which demonstrate significant advantages of FedPepTAO in terms of accuracy (up to 60.8% higher) and efficiency (up to 97.59% faster).

## Acknowledgements

This research is partially sponsored by the National Science Foundation (NSF) under Grant No. OAC-2313191.

## Limitations

While our method can significantly enhance the performance and efficiency of federated prompt tuning in large language models (LLMs), we should acknowledge the sharing of prompts between clients and the server. Previous research has demonstrated that transferring additional sensitive information in Federated Learning (FL), such as predictions or embeddings, can lead to potential privacy concerns (Che et al., 2022). These concerns can be even more critical in prompt tuning scenarios, as prompts are explicitly tuned with private local data. We anticipate that evaluating and mitigating privacy risks in federated LLM prompt tuning will be an intriguing research area in the future.

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

## A Implementation Details

### A.1 Adam Update

The original Adam update (Kingma and Ba, 2015) is shown in Algorithm 3.

---

**Algorithm 3** Adam update

**Require:**
  $w^{t-1}$: The prompt parameters of the model at Iteration $t-1$
  $m^{t-1}$: The $1^{st}$ momentum vector at Iteration $t-1$
  $v^{t-1}$: The $2^{st}$ momentum vector at Iteration $t-1$
  $g^t$: The gradients corresponding to $w^{t-1}$
  $\beta_1, \beta_2$: Decay rates for the momentum in Adam
  $\alpha$: The step size
**Ensure:**
  $w^t$: The prompt parameters at Iteration $t$
  $m^t$: The $1^{st}$ momentum vector at Iteration $t$
  $v^t$: The $2^{st}$ momentum vector at Iteration $t$
1: $m^t \leftarrow \beta_1 m^{t-1} + (1-\beta_1)g^t$
2: $\hat{m}^t \leftarrow \frac{m^t}{1-\beta_1^t}$
3: $v^t \leftarrow \beta_2 v^{t-1} + (1-\beta_2)(g^t)^2$
4: $\hat{v}^t \leftarrow \frac{v^t}{1-\beta_2^t}$
5: $w^t \leftarrow w^{t-1} - \alpha \frac{\hat{m}^t}{(\sqrt{\hat{v}^t}+\epsilon)}$

---

### A.2 Details for Experimental Setup

The number of global training epochs is set to 100 and that of local training epochs is set to 2. We utilize the Dirichlet distribution (with 1.0 as the concentration parameter alpha) to partition the data into non-IID splits and assign a certain number of samples to each device according to the Dirichlet distribution (with 5.0 as the concentration parameter alpha). We exploit development sets for the evaluation of tasks in the GLUE benchmark since test sets are not labeled. For 4 other datasets, we select a certain number of samples from the training set as the development set, and the number of samples for each label is determined according to its proportion in the original training set. For datasets in GLUE benchmark (Wang et al., 2019), we use their original data splits. For 4 other datasets with no default splits, we randomly divide the dataset into train, development, and test sets. The dataset statistics after the split are shown in Table 13

We set prompt lengths for each method according to the original works, i.e., 128 for FedPrompt and P-tuning v2, 5 for LPT and IDPG, 100 for AT-TEMPT, and 20 for Prompt tuning. FedPrompt, Prompt tuning, and ATTEMPT insert prompts before the transformer layers. P-tuning v2 inserts prompts to the hidden states for all transformer layers. IDPG combines these two heuristics and inserts prompts to either the input or the hidden states of all layers. LPT searches for the single best layer by training all the possible positions for each layer. Similar to P-tuning v2, FedPepTAO inserts the hidden states for all transformer layers while the prompt parameters of properly selected layers are communicated between devices and the server.

## B Extra Experiments

### B.1 Epochs Required to Achieve the Target Accuracy

We conducted experiments with the number of epochs required to achieve the target accuracy and the communication overhead to demonstrate the performance of FedPepTAO on the RoBERTa$_{LARGE}$ model and 10 tasks. Below are the average epochs required to achieve the target accuracy.

| Method | Epochs |
|---|---|
| Adapter | 30.92 |
| FedPrompt | 24.50 |
| P-tuning v2 | 5.99 |
| Prompt Tuning | / |
| IDPG | 40.15 |
| ATTEMPT | 39.97 |
| LPT | 8.31 |
| MomD | 13.98 |
| MomS+AdamD | 5.99 |
| FedPepTAO | 3.88 |

Table 6: The average number of epochs required to achieve the target accuracy (85% for QNLI, 92.5% for SST-2, 3% for CoLA, 77% for MRPC, 65% for RTE, 71% for BoolQ, 85% for MPQA, 88% for Subj, 78% for Trec, 91% for MR) on RoBERTa$_{LARGE}$ model. "/" represents that training does not achieve the target accuracy.

From Table 6, we find that FedPepTAO requires the smallest amount of epochs to achieve the target accuracy (87.45%, 84.16%, 35.26%, 90.33%, 90.29%, 53.31%, 72.22%, 35.21% faster compared to Adapter, FedPrompt, P-tuning v2, IDPG, AT-TEMPT, LPT, MomD, and MomS+AdamD respectively). Prompt Tuning failed to achieve the target accuracy since it only optimizes the first layer of soft prompts. FedPepTAO can also reduce the

communication overhead from 40% to 41.55% (41.55%, 40%, 40%, 40% compared with Adapter, P-tuning v2, MomD, and MomS+AdamD, respectively) between devices and the server, as illustrated in Table 7. FedPrompt, Prompt-Tuning, IDPG, ATTEMPT, and LPT correspond to lower communication overhead (up to 13%) since they only select one layer during tuning, which results in significantly inferior accuracy (from 17.02% to 60.8% lower) compared with FedPepTAO.

| Method | Time |
| --- | --- |
| Adapter | 5.80 |
| FedPrompt | 3.09 |
| P-tuning v2 | 5.65 |
| Prompt Tuning | 3.00 |
| IDPG | 3.09 |
| ATTEMPT | 3.16 |
| LPT | 3.09 |
| MomD | 5.65 |
| MomS+AdamD | 5.65 |
| FedPepTAO | 3.39 |

Table 7: The communication overhead (s) between devices and the server with RoBERTa$_{LARGE}$ model.

## B.2 Impact of server learning rate

Due to the high sensitivity of the hyperparameters in NLP tasks, we investigate the impact of divers server learning rates on the RTE dataset. We analyze the accuracy with the learning rates $1e^{-2}$, $5e^{-3}$, $1e^{-3}$, $5e^{-4}$. As shown in Figure 2, we find that the best performance was achieved when $lr = 1e^{-3}$, which only slightly outperforms $lr = 5e^{-4}$ by 0.37%. This demonstrates that FedPepTAO is easy to fine-tune in practice.

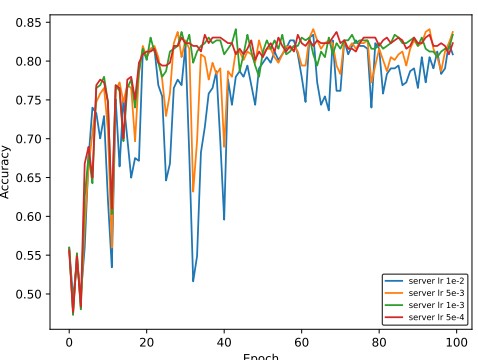

Figure 2: The impact of server learning rate.

## B.3 Impact of heterogeneous data distribution

To investigate the robustness of our approach to different degrees of non-IID data, we conduct experiments under various levels of non-IID degrees. From Figure 3, we observe that the accuracy of FedPepTAO is the best with $\alpha$ ranging from 1.0 to 5.0. A smaller $\alpha$ represents a higher degree of heterogeneity. This demonstrates that our approach is robust to different data heterogeneity.

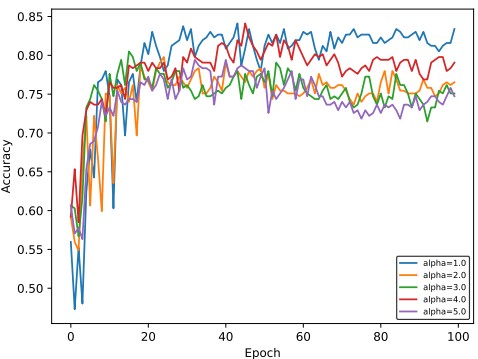

Figure 3: The impact of various heterogeneity degrees.

## B.4 Impact of device number

To explore the scalability of our model, we conduct experiments with divers number of devices, i.e., 100, 150, and 200. Figure 4 shows the corresponding accuracy on the RTE dataset. The accuracy gap between the best and worst is only 0.73%, which demonstrates that FedPepTAO is scalable in FL settings.

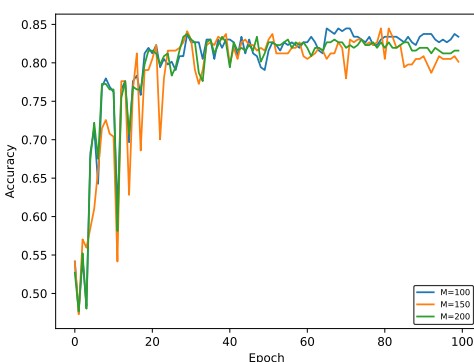

Figure 4: Evaluation of divers number of devices.

## B.5 Impact of diverse bandwidth

We carry out extra experimentation on two tasks, i.e., Subj and Trec with modest network bandwidth (reduced to 100 times smaller) on the RoBERTa$_{LARGE}$ model. We find that FedPepTAO maintains its advantages in this setting, i.e., up to 98.71%, 94.75%, 80.61%, 95%, 97.43%, 56.52%, 84.55% faster compared with Adapter, FedPrompt, P-tuning v2, IDPG, ATTEMPT, LPT, MomS+AdamD to achieve the target accuracy, as shown in Table 8.

| Method | Subj | Trec |
|---|---|---|
| Adapter | 3591 | 7908 |
| FedPrompt | 1333 | 368 |
| P-tuning v2 | 361 | 474 |
| Promtp Tuning | / | / |
| IDPG | 1401 | 1345 |
| ATTEMPT | 2726 | 1184 |
| LPT | 161 | 123 |
| MomD | / | / |
| MomS+AdamD | 453 | 496 |
| FedPepTAO | 70 | 102 |

Table 8: The tuning time (s) to achieve a target accuracy (88% for Subj, 78% for Trec) on RoBERTa$_{LARGE}$ model. "/" represents that training does not achieve the target accuracy.

## B.6 Parameter-efficient Prompt Tuning and random layer selection strategy

In order to clarify the impact of randomness in our experiments, we conduct three experiments with random layer selection strategy on RTE dataset. As shown in Table 9, FedPepTAO outperforms the random strategy with the accuracy gain of 2.53%, 2.89%, and 3.25% respectively, which demonstrates the superior performance of our FedPepTAO method.

| Method | Seed 42 Acc |
|---|---|
| FedPepTAO | 85.56% |
| Random 1 | 83.03% |
| Random 2 | 82.67% |
| Random 3 | 82.31% |
| Avg acc gain | 2.89% |

Table 9: The performance of FedPepTAO and random layer choosing strategy on RTE dataset.

In addition, in order to further validate the impact of randomness on different datasets, we conducted additional experiments on three datasets (RTE, MRPC, and CoLA) with three randomly selected seeds (32, 35, and 37) to testify the strength of our Parameter-efficient Prompt Tuning method.

Tables 10, 11, and 12 exhibit that FedPepTAO outperforms the random strategy by 2.91%, 4.64%, and 7.29% on RTE, MRPC, and CoLA datasets, respectively. The above experiment results indicate that our FedPepTAO method can achieve substantial improvement compared with the random strategy.

| Method | Seed 32 Acc | Seed 35 Acc | Seed 37 Acc |
|---|---|---|---|
| FedPepTAO | 84.84% | 85.56% | 85.92% |
| Random 1 | 82.31% | 82.31% | 82.31% |
| Random 2 | 81.59% | 81.23% | 83.39% |
| Random 3 | 82.67% | 83.03% | 83.75% |
| Avg Acc Gain | 2.65% | 3.37% | 2.77% |

Table 10: The performance of FedPepTAO and random layer choosing strategy on RTE dataset under different random seeds

| Method | Seed 32 Acc | Seed 35 Acc | Seed 37 Acc |
|---|---|---|---|
| FedPepTAO | 86.54% | 87.09% | 86.18% |
| Random 1 | 83.09% | 81.86% | 80.34% |
| Random 2 | 82.73% | 82.03% | 80.54% |
| Random 3 | 83.31% | 81.83% | 81.16% |
| Avg Acc Gain | 3.5% | 5.18% | 5.5% |

Table 11: The performance of FedPepTAO and random layer choosing strategy on MRPC dataset under different random seeds

| Method | Seed 32 Acc | Seed 35 Acc | Seed 37 Acc |
|---|---|---|---|
| FedPepTAO | 56.94% | 58.92% | 56.49% |
| Random 1 | 46.45% | 52.84% | 49.58% |
| Random 2 | 51.57% | 49.84% | 45.71% |
| Random 3 | 50.77% | 53.2% | 51.45% |
| Avg Acc Gain | 7.34% | 6.96% | 7.58% |

Table 12: The performance of FedPepTAO and random layer choosing strategy on CoLA dataset under different random seeds

We notice an inverse correlation between the performance of our Parameter-efficient Prompt Tuning (PEPT) method and the average sentence length of the three datasets. Specifically, PEPT tends to achieve a smaller performance gain on the datasets with longer average sentence length, as shown in Table 14.

A reasonable explanation is that the datasets with longer average sentence length, such as RTE, often contain more latent information. More/less latent information make them easier/more difficult to be

| Category | Datasets | \|**Train**\| | \|**Dev**\| | \|**Test**\| | $\|\mathcal{Y}\|$ | Type | Labels |
|---|---|---|---|---|---|---|---|
| | SST-2 | 67349 | 872 | 1821 | 2 | sentiment | positive, negative |
| | MPQA | 7606 | 1000 | 2000 | 2 | opinion polarity | positive, negative |
| Single-sentence | MR | 7662 | 1000 | 2000 | 2 | sentiment | positive, negative |
| | Subj | 7000 | 1000 | 2000 | 2 | subjectivity | subjective, objective |
| | Trec | 4952 | 500 | 500 | 6 | question cls. | abbr., entity, description, human, loc., num. |
| | CoLA | 8551 | 1043 | 1063 | 2 | acceptability | acceptable, unacceptable |
| | MRPC | 3668 | 408 | 1725 | 2 | paraphrase | equivalent, not equivalent |
| | QNLI | 104743 | 5463 | 5463 | 2 | NLI | entailment, not entailment |
| Sentence-pair | BoolQ | 9427 | 3270 | 3245 | 2 | QA | true, false |
| | RTE | 2490 | 277 | 3000 | 2 | NLI | entailment, not entailment |

Table 13: The statistics of datasets evaluated in this work. $\|\mathcal{Y}\|$ is the number for classes.

| Dataset | Avg sentence length | Avg acc gain |
|---|---|---|
| RTE | 71.91 | 2.91% |
| MRPC | 54.95 | 4.64% |
| CoLA | 16.37 | 7.29% |

Table 14: Average sentence length and accuracy improvements of FedPepTAO for three datasets.

evenly distributed across all transformer layers, resulting in a relatively equal/diverse contribution by each layer during tuning. When different layers contain the similar/dissimilar amount of latent information, the impact of each unique layer is accordingly decreased/increased. Therefore, the random layer selection results in less/more accuracy gain by our PEPT method. The above experiment results demonstrate that our PEPT method can achieve substantial performance improvement on different datasets in most experiments.

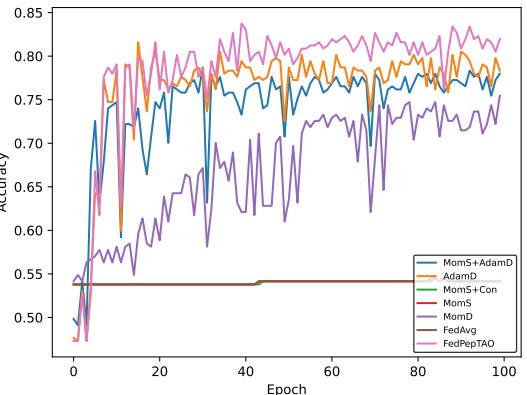

Figure 6: Evaluation for various optimization methods.

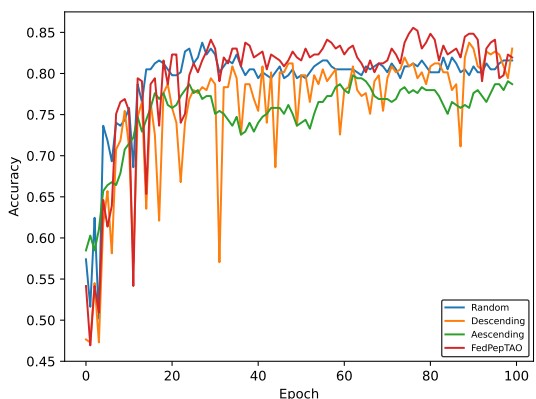

Figure 5: Evaluation of divers layer selection strategies.