# OpenReview forum: "Federated Learning of Large Language Models with Parameter-Efficient Prompt Tuning and Adaptive Optimization"
_EMNLP/2023/Conference — EMNLP 2023 Main_

### Official Review · Reviewer_tTk3 · 2023-08-04

**Soundness:** 4

**Excitement:**

4: Strong: This paper deepens the understanding of some phenomenon or lowers the barriers to an existing research direction.

**Paper Topic And Main Contributions:**

Federated Learning (FL) is a novel idea for collaborative model training using decentralized data. However, the training process of Large Language Models (LLMs) often involves the updating of major parameters, limiting the application of FL approaches to LLMs in real-world scenarios. Prompt tuning could result in outstanding performance while freezing the original LLMs, via searching proper prompts in the discrete space.  However, existing approaches for fine-tuning LLMs significantly degrade performance due to the massive amount of communication.
Authors propose FedPepTAO, a Parameter-Efficient prompt Tuning technique with adaptive optimization, to enable efficient and effective FL of LLMs. First, an effective partial prompt tuning strategy is given to increase both performance and efficiency by selecting appropriate layers of prompts for FL. A subset of layers of prompts can minimize both communication and processing costs. Second, to improve performance even further, a novel adaptive optimization strategy is created to handle client drift issues on both the device and server sides.
Experiments show that  FedPepTAO outperforms 9 baselines on 10 benchmark datasets.

**Questions For The Authors:**

1) Please provide an evaluation of runtime performance on different hardware environments, such as various network bandwidth. Because network communication and on-device computing are overlapping, the metric of communication parameters simply reflects network cost and does not show system efficiency. The tuning time is heavily dependent on network bandwidth.

2)  what is the step size of learning rate from $1e^{-2}$ to $5e^{-4}$?

3) no Appendix in this submission version.

4) Authors utilize batch data on the hidden states of each layer to select the proper layers. If randomly choosing the same number of subset layers, removing the layer scoring stage, how about the performance of FedPepTAO?

5) How much the component of communication adaptive optimization contributes in the final performance?

**Reasons To Accept:**

1) Authors propose a new Prompt Tuning, instead of tuning the full set of prompts parameters, selects a part of layers depending on their importance score for trading prompt parameters during tuning process.
2) Authors use an adaptive optimization to capture non-IID data to avoid extra data transmission and reduce communication cost.

**Reasons To Reject:**

1) (If the title includes the phrase "Large Language Models"), the experiments were carried out using the RoBERTa-large model with only 355M parameters, which is far from LLMs, which are typically at least 7B in size.

**Reproducibility:**

3: Could reproduce the results with some difficulty. The settings of parameters are underspecified or subjectively determined; the training/evaluation data are not widely available.

**Reviewer Confidence:**

3: Pretty sure, but there's a chance I missed something. Although I have a good feel for this area in general, I did not carefully check the paper's details, e.g., the math, experimental design, or novelty.

---

> ### Author Rebuttal · Authors · 2023-08-29
>
> We sincerely thank the reviewer for valuable comments. We use RR.n and Q.n to denote the response to “Reasons To Reject,” or ``Questions For The Authors” n. For the reviewers’ convenience, we retype the review comments and suggestions in bold followed by our responses. The response will be clarified in the revised paper.
>
> **RR.1: (If the title includes the phrase "Large Language Models"), the experiments were carried out using the RoBERTa-large model with only 355M parameters, which is far from LLMs, which are typically at least 7B in size.**
>
> RR.1 Response: Thanks for the valuable comment. To demonstrate the adaptability of FedPepTAO, we carried out extra experiments with three additional decoder-based models, i.e., GPT-2 model (774M) on MRPC, MR, SST-2 dataset, LLaMA 3B model on RTE, MRPC dataset, and LLaMA 7B model on MRPC dataset.
>
> As shown in Table 1 below, FedPepTAO significantly outperforms baseline methods in terms of the best accuracy on the decoder-based GPT-2 model (up to 32.3\%, 18.23\%, 43.32\%, 15.94\%, 4\%, 4.34\% higher compared to FedPrompt, P-tuning v2, ATTEMPT, LPT, MomD and MomS+AdamD, respectively). Furthermore, the efficiency of FedPepTAO is significantly higher than baseline approaches (up to 84.74\%, 86.89\%, and 15.27\% faster compared to LPT, MomD, and MomS+AdamD, respectively).
>
> Table 1. Accuracy and tuning time (s) to achieve target accuracy (75% for MRPC, 81% for MR, and 92.5% for SST-2) on GPT-2 model. "/" represents that training does not achieve the target accuracy.
> | Method          | MRPC Acc | MRPC Time | MR Acc | MR Time | SST-2 Acc | SST-2 Time |
> |-----------------|---------|-----------|--------|---------|-----------|------------|
> | FedPrompt       | 74.98   |     /     | 57.2   |    /    | 76.49     |     /      |
> | P-tuning v2     | 74.8    |     /     | 73     |    /    | 74.77     |     /      |
> | ATTEMPT         | 37.91   |     /     | 57.3   |    /    | 85.89     |     /      |
> | LPT             | 74.98   |     /     | 84.8   |  1455   | 77.06     |     /      |
> | MomD            | 77.23   |   503     | 85.7   |  1694   | 92.55     |    3638    |
> | MomS+AdamD      | 76.89   |   291     | 88.2   |   262   | 92.55     |    2488    |
> | FedPepTAO       | 81.23   |   273     | 89.5   |   222   | 93        |    2248    |
>
> When the model becomes larger, i.e., LLaMA 3B with 3 billion parameters, FedPepTAO still achieves the best accuracy (up to 4.6\%, 11.29\%, 5.28\%, 6.69\%, 6.2\%, 10.88\% higher compared to FedPrompt, P-tuning v2, ATTEMPT, LPT, MomD and MomS+AdamD, respectively) and better efficiency (up to 39.81\%, 43.04\%, 48.16\%, 38.86\%, 9.72\% faster compared to FedPrompt, ATTEMPT, LPT, MomD, and MomS+AdamD, respectively) as illustrated in Table 2.
>
> Table 2. Accuracy and tuning time (s) to achieve target accuracy (75% for RTE, 81% for MRPC) on LLaMA 3B model. "/" represents that training does not achieve the target accuracy.
> | Method        | RTE Acc | RTE Time | MRPC Acc | MRPC Time |
> |---------------|---------|----------|----------|----------|
> | FedPrompt     | 78.34   | 540      | 81.86    | 459      |
> | P-tuning v2   | 56.68   |   /      | 75.17    |   /      |
> | ATTEMPT       | 64.98   |   /      | 81.18    | 718      |
> | LPT           | 64.98   |   /      | 79.77    | 789      |
> | MomD          | 80.87   | 360      | 80.26    | 669      |
> | MomS+AdamD    | 80.87   | 360      | 75.58    |   /      |
> | FedPepTAO     | 83.39   | 325      | 86.46    | 409      |
>
>
> We further verify the performance of our method with another large model, i.e., LLaMA 7B with 7 billion parameters. As presented in Table 3, FedPepTAO outperforms baseline methods in terms of accuracy (up to 5.05\%, 26.71\%, 18.41\%, 18.41\%, 6.2\%, 10.88\% higher compared to FedPrompt, P-tuning v2, ATTEMPT, LPT, MomD and MomS+AdamD, respectively) and efficiency (up to 15.77\%, 75.14\%, 32.06\%, 67.04\% faster compared to ATTEMPT, LPT, MomD, and MomS+AdamD, respectively), which demonstrates the scalability of FedPepTAO.
>
> Table 3. Accuracy and tuning time (s) to achieve target accuracy (75% for MRPC) on LLaMA 7B model. "/" represents that training does not achieve the target accuracy.
> | Method       | MRPC Acc | MRPC Time |
> |--------------|----------|-----------|
> | FedPrompt    | 76.18    |    /      |
> | P-tuning v2  | 74.98    |    /      |
> | ATTEMPT      | 81.52    |   317     |
> | LPT          | 81.95    |  1074     |
> | MomD         | 76.2     |   393     |
> | MomS+AdamD   | 75.75    |   810     |
> | FedPepTAO    | 82.34    |   267     |
>
> This will be clarified in the revised version.
>
> **Q.1: Please provide an evaluation of runtime performance on different hardware environments, such as various network bandwidth. Because network communication and on-device computing are overlapping, the metric of communication parameters simply reflects network cost and does not show system efficiency. The tuning time is heavily dependent on network bandwidth.**
>
> Q.1 Response: Thanks for the thoughtful suggestion. We carried out extra experimentation on two tasks, i.e., Subj and Trec with modest network bandwidth (reduced to 100 times smaller) on RoBERTA-Large model. We find that FedPepTAO maintains its advantages in this setting, i.e., up to 98.71\%, 94.75\%, 80.61\%, 95\%, 97.43\%, 56.52\%, 84.55\% faster compared with Adapter, FedPrompt, P-tuning v2, IDPG, ATTEMPT, LPT, MomS+AdamD  to achieve the target accuracy, as shown in Table 4 below. This will be clarified in the revised version.
>
> Table 4. The tuning time (s) to achieve a target accuracy (88% for Subj, 78% for Trec) on RoBERTA-Large model. "/" represents that training does not achieve the target accuracy.
> | Method        | Subj   | Trec   |
> |---------------|--------|--------|
> | Adapter       | 3591   | 7908   |
> | FedPrompt     | 1333   | 368    |
> | P-tuning v2    | 361    | 474    |
> | Promtp Tuning | /      | /     |
> | IDPG          | 1401   | 1345   |
> | ATTEMPT       | 2726   | 1184   |
> | LPT           | 161    | 123    |
> | MomD          | /      | /     |
> | MomS+AdamD    | 453    | 496    |
> | FedPepTAO    | 70     | 102    |
>
> **Q.2: What is the step size of learning rate from $1e^{-2}$ to $5e^{-4}$?**
>
> Q.2 Response: We apologize for not being specific in our evaluation. The step size of learning rate from $1e^{-2}$ to $5e^{-4}$ is $1e^{-2}$, $5e^{-3}$, $1e^{-3}$, and $5e^{-4}$, as illustrated in Figure 5 from Appendix. This will be clarified in the revised version.
>
> **Q.3: no Appendix in this submission version.**
>
> Q.3 Response: Thanks for the comment. The Appendix has been included in the supplementary materials upon the submission deadline in June. Please kindly unzip the supplementary file and find our Appendix.
>
> **Q.4: Authors utilize batch data on the hidden states of each layer to select the proper layers. If randomly choosing the same number of subset layers, removing the layer scoring stage, how about the performance of FedPepTAO?**
>
> Q.4 Response: Thanks for the thoughtful comment. We experimented with various layer selection methods as shown in Figure 2 of the Appendix. From Figure 2 of the paper, we can see that FedPepTAO corresponds to significantly higher accuracy compared with other methods (up to 1.81\% compared with random, up to 1.81\% compared with Descending, and 5.78\% compared with Aescending). This will be clarified in the revised version.
>
> **Q.5: How much the component of communication adaptive optimization contributes in the final performance?**
>
> Q.5 Response: Thanks for the valuable question. As we explained in Section 4.4., the component of communication adaptive optimization can outperform baseline approaches from 2.16\% to 29.24\% (up to 8.3\% compared with MomD, 29.24\% compared with MomS, 29.24\% compared with MomS+Con, 2.16\% compared with AdamD, 5.05\% compared with MomS+AdamD, and 29.24\% compared with FedAvg). In addition, we conducted extra experiments to compare FedPepTAO and FedPepTAO-opt (FedPepTAO without the communication adaptive optimization), where FedPepTAO outperforms FedPepTAO-opt up to 4.69\% in terms of accuracy, which demonstrates the contribution of communication adaptive optimization in the final performance. This will be clarified in the revised version.

---

### Official Review · Reviewer_ZSxS · 2023-08-04

**Soundness:** 3

**Excitement:**

4: Strong: This paper deepens the understanding of some phenomenon or lowers the barriers to an existing research direction.

**Paper Topic And Main Contributions:**

This paper introduces a novel approach called "FedPepTAO" that aims to enable efficient and effective federated learning of LLMs in scenarios where data is decentralized.  This paper addresses is the difficulty of applying federated learning to Large Language Models due to the large number of parameters that need to be updated during training.

Initial solution that authors propose using prompt tuning to reduce the number of parameters, but this straightforward application of prompt tuning in FL leads to high communication costs and performance degradation. Additionally, the decentralized data in FL is often non-Independent and Identically Distributed (non-IID), leading to client drift problems and poor performance.

I think it’s possible to summarize the paper contributions in 2 directions: Partial prompt tuning and Adaptive Optimization for Client Drift.

The authors also provided results on 10 datasets to evaluate the performance of FedPepTAO. The results demonstrate significant improvements in terms of accuracy (up to 60.8%) and training time efficiency (up to 97.59%) compared to 9 baseline approaches.


**Questions For The Authors:**


A: Can you give the logic behind calculating the Hessian Matrix? Are there any alternative approaches?
B: Assuming lines 3 - 14 happen in parallel, during the aggregation stage what is the expected communication overhead? You mentioned that aggregating prompt parameter ratio, but this is not defined as far as I can see. I assume this is something negligible.
C: Can you explain how maintaining a state for each device on the server avoids possible drift problem? Do you have any empirical data on how effective this is?


**Reasons To Accept:**

I found the both contributions useful to the community.

* Efficient Partial Prompt Tuning: The paper introduces an efficient partial prompt tuning approach that improves performance and efficiency simultaneously. This approach helps to reduce the number of parameters that need to be updated during FL, making it more feasible for LLMs.
* Adaptive Optimization for Client Drift: A novel adaptive optimization method is proposed to address client drift problems on both the device and server sides, further enhancing the performance of FL for LLMs. This addresses the challenges posed by non-IID data distribution in decentralized FL settings.

 The authors approach is technically sound, and the description of the approach seems implementable.


**Reasons To Reject:**

My biggest concern is how the approach scale with larger models like GLM XLarge GLM XXLarge. I can understand the limitations on the experimental side, however, it would be great to hear from authors regarding how the performance when the model gets larger. Is this approach still feasible? What are some disadvantages that you foresee?


**Reproducibility:**

4: Could mostly reproduce the results, but there may be some variation because of sample variance or minor variations in their interpretation of the protocol or method.

**Reviewer Confidence:**

3: Pretty sure, but there's a chance I missed something. Although I have a good feel for this area in general, I did not carefully check the paper's details, e.g., the math, experimental design, or novelty.

**Typos Grammar Style And Presentation Improvements:**

I think FedAvg should be explained in the paper. It’s hard for the reader to follow up on that.
I think paper reads well, so thanks for your efforts.

---

> ### Author Rebuttal · Authors · 2023-08-29
>
> We sincerely thank the reviewer for valuable comments. We use RR.n and Q.n to denote the response to “Reasons To Reject,” or ``Questions For The Authors” n. For the reviewers’ convenience, we retype the review comments and suggestions in bold followed by our responses. The response will be clarified in the revised paper.
>
> **RR.1: My biggest concern is how the approach scale with larger models like GLM XLarge GLM XXLarge. I can understand the limitations on the experimental side, however, it would be great to hear from authors regarding how the performance when the model gets larger. Is this approach still feasible? What are some disadvantages that you foresee?**
>
> RR.1 Response: Thanks for the valuable comment. This approach is still feasible. FedPepTAO can reduce the number of parameters to transfer with our parameter-efficient prompt tuning method and perform adaptive optimization without extra communication costs, leading to better efficiency.
>
> To demonstrate the scalability of FedPepTAO, we carried out extra experiments with three additional decoder-based models, i.e., GPT-2 model (774M) on MRPC, MR, SST-2 dataset, LLaMA 3B model on RTE, MRPC dataset, and LLaMA 7B model on MRPC dataset. As shown in Table 1 below, FedPepTAO significantly outperforms baseline methods in terms of the best accuracy on the decoder-based GPT-2 model (up to 32.3\%, 18.23\%, 43.32\%, 15.94\%, 4\%, 4.34\% higher compared to FedPrompt, P-tuning v2, ATTEMPT, LPT, MomD and MomS+AdamD, respectively). Furthermore, the efficiency of FedPepTAO is significantly higher than baseline approaches (up to 84.74\%, 86.89\%, and 15.27\% faster compared to LPT, MomD, and MomS+AdamD, respectively).
>
> Table 1. Accuracy and tuning time (s) to achieve target accuracy (75% for MRPC, 81% for MR, and 92.5% for SST-2) on GPT-2 model. "/" represents that training does not achieve the target accuracy.
> | Method          | MRPC Acc | MRPC Time | MR Acc | MR Time | SST-2 Acc | SST-2 Time |
> |-----------------|---------|-----------|--------|---------|-----------|------------|
> | FedPrompt       | 74.98   |     /     | 57.2   |    /    | 76.49     |     /      |
> | P-tuning v2     | 74.8    |     /     | 73     |    /    | 74.77     |     /      |
> | ATTEMPT         | 37.91   |     /     | 57.3   |    /    | 85.89     |     /      |
> | LPT             | 74.98   |     /     | 84.8   |  1455   | 77.06     |     /      |
> | MomD            | 77.23   |   503     | 85.7   |  1694   | 92.55     |    3638    |
> | MomS+AdamD      | 76.89   |   291     | 88.2   |   262   | 92.55     |    2488    |
> | FedPepTAO       | 81.23   |   273     | 89.5   |   222   | 93        |    2248    |
>
> When the model becomes larger, i.e., LLaMA 3B with 3 billion parameters, FedPepTAO still achieves the best accuracy (up to 4.6\%, 11.29\%, 5.28\%, 6.69\%, 6.2\%, 10.88\% higher compared to FedPrompt, P-tuning v2, ATTEMPT, LPT, MomD and MomS+AdamD, respectively) and better efficiency (up to 39.81\%, 43.04\%, 48.16\%, 38.86\%, 9.72\% faster compared to FedPrompt, ATTEMPT, LPT, MomD, and MomS+AdamD, respectively) as illustrated in Table 2.
>
> Table 2. Accuracy and tuning time (s) to achieve target accuracy (75% for RTE, 81% for MRPC) on LLaMA 3B model. "/" represents that training does not achieve the target accuracy.
> | Method        | RTE Acc | RTE Time | MRPC Acc | MRPC Time |
> |---------------|---------|----------|----------|----------|
> | FedPrompt     | 78.34   | 540      | 81.86    | 459      |
> | P-tuning v2   | 56.68   |   /      | 75.17    |   /      |
> | ATTEMPT       | 64.98   |   /      | 81.18    | 718      |
> | LPT           | 64.98   |   /      | 79.77    | 789      |
> | MomD          | 80.87   | 360      | 80.26    | 669      |
> | MomS+AdamD    | 80.87   | 360      | 75.58    |   /      |
> | FedPepTAO     | 83.39   | 325      | 86.46    | 409      |
>
>
> We further verify the performance of our method with another large model, i.e., LLaMA 7B with 7 billion parameters. As presented in Table 3, FedPepTAO outperforms baseline methods in terms of accuracy (up to 5.05\%, 26.71\%, 18.41\%, 18.41\%, 6.2\%, 10.88\% higher compared to FedPrompt, P-tuning v2, ATTEMPT, LPT, MomD and MomS+AdamD, respectively) and efficiency (up to 15.77\%, 75.14\%, 32.06\%, 67.04\% faster compared to ATTEMPT, LPT, MomD, and MomS+AdamD, respectively), which demonstrates the scalability of FedPepTAO.
>
> Table 3. Accuracy and tuning time (s) to achieve target accuracy (75% for MRPC) on LLaMA 7B model. "/" represents that training does not achieve the target accuracy.
> | Method       | MRPC Acc | MRPC Time |
> |--------------|----------|-----------|
> | FedPrompt    | 76.18    |    /      |
> | P-tuning v2  | 74.98    |    /      |
> | ATTEMPT      | 81.52    |   317     |
> | LPT          | 81.95    |  1074     |
> | MomD         | 76.2     |   393     |
> | MomS+AdamD   | 75.75    |   810     |
> | FedPepTAO    | 82.34    |   267     |
>
> This will be clarified in the revised version.
>
> **Q.A: Can you give the logic behind calculating the Hessian Matrix? Are there any alternative approaches?**
>
> Q.A Response: Thanks for the thoughtful question. We exploit an efficient PyHessian library [1] to calculate the eigenvalues in the Hessian Matrix. With the PyHessian library, we can calculate the trace and the Eigenvalue Spectral Density (ESD) of the Hessian matrix. With the trace and ESD, we can calculate each eigenvalue of the Hessian matrix. Within PyHessian, the trace is calculated using randomized numerical linear algebra with Hutchinson's method for fast computation with Hessian matvec computations (see [1] for details). In addition, PyHessian computes the ESD with 4 steps. First, it approximates the ESD with a Gaussian kernel. Then, it simplifies the problem with Gaussian quadrature. Afterward, it exploits the stochastic Lanczos algorithm to approximate the parameters in the ESD. Finally, it approximates the expectation with multiple times of the Lanczos algorithm execution.
>
> There are other approaches to calculate the eigenvalues of the Hessian Matrix [4, 5], or ESD [2, 3, 6]. However, there is no open-source library to exploit for [2-6] while PyHessian can be directly utilized via [7]. The Hessian matrix can be directly calculated with TORCH.AUTOGRAD.FUNCTIONAL.HESSIAN [8] in PyTorch. However, the HESSIAN function in PyTorch (180.9 seconds) incurs significantly longer time (35 times longer) compared with PyHessian (4.8 seconds), based on a simple experiment with a deep neural network of three Multi-Layer Perceptron layers and 8090 neurons. In addition, the relative error of PyHessian is less than 5\%, which is acceptable. Furthermore, the HESSIAN function in PyTorch (2.64GB) requires 4 times bigger memory compared with PyHessian (528MB). As a result, the HESSIAN function in PyTorch does not work in our case due to unaccepted execution time and out-of-memory issues with big models, and we utilize PyHessian. This will be clarified in the revised version.
>
>
>
> [1] Z. Yao, A. Gholami, K. Keutzer, and M. W. Mahoney. 2020. Pyhessian: Neural networks through the lens of the hessian. In IEEE Int. Conf. on Big Data (Big Data), pages 581–590. IEEE Computer Society.
>
> [2] Lin Lin, Yousef Saad, and Chao Yang. Approximating spectral densities of large matrices. SIAM review,
> 58(1):34–65, 2016.
>
> [3] Shashanka Ubaru, Jie Chen, and Yousef Saad. Fast estimation of tr(f(a)) via stochastic Lanczos quadrature. SIAM Journal on Matrix Analysis and Applications, 38(4):1075–1099, 2017.
>
> [4] Levent Sagun, Leon Bottou, and Yann LeCun. Eigenvalues of the Hessian in deep learning: Singularity and beyond. arXiv preprint arXiv:1611.07476, 2016.
>
> [5] Levent Sagun, Utku Evci, V Ugur Guney, Yann Dauphin, and Leon Bottou. Empirical analysis of the hessian of over-parametrized neural networks. arXiv preprint arXiv:1706.04454, 2017.
>
> [6] Behrooz Ghorbani, Shankar Krishnan, and Ying Xiao. An investigation into neural net optimization via Hessian eigenvalue density. arXiv preprint arXiv:1901.10159, 2019.
>
> [7] PyHessian: https://github.com/amirgholami/PyHessian
>
> [8] https://pytorch.org/docs/stable/generated/torch.autograd.functional.hessian.html
>
> **Q.B: Assuming lines 3 - 14 happen in parallel, during the aggregation stage what is the expected communication overhead? You mentioned that aggregating prompt parameter ratio, but this is not defined as far as I can see. I assume this is something negligible.**
>
> Q.B Response: Thanks for your helpful comment. Yes, the communication overhead is negligible. The communication overhead of the aggregation stage for each device is composed of two parts. The first part is the ratio between the selected layers and the whole layer set, which is a float value with 4 bytes. The second part is the score of each layer on each device, which is a set of $L$ float values with 4*L bytes. The total communication overhead is 4 *(L + 1) bytes, which are negligible (less than 1K bytes). The prompt parameter ratio refers to the value of $r_i$ in Line 10, which is the first part of the communication overhead. This will be clarified in the revised version.
>
> **Q.C: Can you explain how maintaining a state for each device on the server avoids possible drift problem? Do you have any empirical data on how effective this is?**
>
> Q.C Response: Thank you for the insightful question. FL as a distributed optimization method has a key challenge: the heterogeneity (non-iid-ness) in the data present on the different clients. Previous research has shown that such heterogeneity introduces a drift in the updates of each client, i.e., client drift, resulting in inferior and unstable convergence [1].
>
> When communication budget is not a concern, the ideal momentum update on device $i$ would be:
>
> $w_i^r = w^{r - 1} - \eta^r * \frac{1}{|M|} \sum_j^{|M|} m_j^r$,
>
> where $\frac{1}{|M|} \sum_j^{|M|} m_j^r$ represents an unbiased mini-batch gradient of $f$. This update is not feasible since it requires communication with all clients for every update step [2, 3]. FedPepTAO instead exploits a state for each device on the server such that
>
> $c_i^r \approx m_i^r$,
>
> $c_{g}^r \approx \frac{1}{|M|} \sum_j^{|M|} m_j^r$,
>
> $\beta*m^{r-1}+(1-\beta)*g_g^r \approx \frac{1}{|S|} \sum_j^{|S|} m_j^r$,
>
> where FedPepTAO can mimic the ideal momentum update with the following formula:
>
> $m_i^r = \beta*m^{r-1}+(1-\beta)*g_g^r + c_{g}^r - c_i^r \approx \frac{1}{|S|} \sum_j^{|S|} m_j^r + \frac{1}{|M|} \sum_j^{|M|} m_j^r - m_i^r \approx \frac{1}{|M|} \sum_j^{|M|} m_j^r$.
>
> And, we have:
>
> $w_i^r = w^{r - 1} - \eta^r * m_i^r = w^{r - 1} - \eta^r * \frac{1}{|M|} \sum_j^{|M|} m_j^r$.
>
> Then, the momentum update of FedPepTAO for each device remains synchronized and converges for the ideal momentum update. In conclusion, we maintain states for each device to overcome gradient dissimilarity, and we maintain them on the server instead of the local devices to avoid additional communication overhead, which is crucial in LLM scenarios.
>
> [1] Sai Praneeth Karimireddy, Satyen Kale, Mehryar Mohri, Sashank Reddi, Sebastian Stich, and Ananda Theertha Suresh. 2020b. Scaffold: Stochastic controlled averaging for federated learning. Int. Conf. on Machine Learning (ICML), pages 5132–5143. PMLR.
>
> [2] Sai Praneeth Karimireddy, Martin Jaggi, Satyen Kale, Mehryar Mohri, Sashank J Reddi, Sebastian U Stich, and Ananda Theertha Suresh. 2020a. Mime: Mimicking centralized stochastic algorithms in federated learning. arXiv preprint arXiv:2008.03606.
>
> [3] Jiayin Jin, Jiaxiang Ren, Yang Zhou, Lingjuan Lyu, Ji Liu, and Dejing Dou. 2022. Accelerated federated learning with decoupled adaptive optimization. In Int. Conf. on Machine Learning (ICML), volume 162 of Proceedings of Machine Learning Research, pages 10298–10322. PMLR.
>
> The experimental results shown in Table 1 of the paper reveal that FedPepTAO achieves significantly higher accuracy (33.21\% for RTE) compared with baseline approaches because of the state of each device with our adaptive optimization method. In addition, we compare the single module of our communication-efficient adaptive optimization with other adaptive optimization baseline approaches without states for each device, which demonstrate significant advantages (up to 29.24\% compared with FedAvg, 8.3\% compared with MomD, 29.24\% compared with MomS, 29.24\% compared with MomS+Con, 2.16\% compared with AdamD, 5.05\% compared with MomS+AdamD higher in terms of accuracy) of our method, as shown in Figure 4. Furthermore, we carried out extra experimentation to show that the value of the state ($c_i^r$) on each device is 35.91\% (on average) compared with that of the momentum ($m_i^r$), which reveals that the state for each device is effective while addressing the client drift problem. This will be clarified in the revised version.

---

### Official Review · Reviewer_nVG8 · 2023-08-05

**Typos Grammar Style And Presentation Improvements:** No severe grammar issue, but writing …
**Soundness:** 2

**Excitement:**

3: Ambivalent: It has merits (e.g., it reports state-of-the-art results, the idea is nice), but there are key weaknesses (e.g., it describes incremental work), and it can significantly benefit from another round of revision. However, I won't object to accepting it if my co-reviewers champion it.

**Missing References:**

N/A

**Paper Topic And Main Contributions:**

This paper proposes parameter-efficient and effective federated learning (FL) of Large language models (LLM). First, an efficient partial prompt tuning approach is proposed to improve performer CSE and efficiency simultaneously. Second a novel adaptive optimization method is developed to address the client drift problems. The exp shows performance improvements for both accuracy and efficiency.

**Questions For The Authors:**

 Table 2, to reach a target acc, reviewer wanna know how those acc are set as target? We know the time to achieve an acc may be vary for different factors. Reviewer believe use a epoch and communication overhead will be more reasonable. How many times did the authors did this experiment? How do authors get those number, the best runtime? Avg runtime?


**Reasons To Accept:**

1. Experimental results seems good.
2. This paper uses matrix decomposition

**Reasons To Reject:**

1. Figure 1 is hard to follow. There is no caption for Figure 1, really can not understand difference between two step 5 and two step 1.
2. Table 2, to reach a target acc, reviewer wanna know how those acc are set as target? We know the time to achieve an acc may be vary for different factors. Reviewer believe use a epoch and communication overhead will be more reasonable. How many times did the authors did this experiment? How do authors get those number, the best runtime? Avg runtime?
3. They only has one model, which sounds not sufficient.

**Reproducibility:**

2: Would be hard pressed to reproduce the results. The contribution depends on data that are simply not available outside the author's institution or consortium; not enough details are provided.

**Reviewer Confidence:**

3: Pretty sure, but there's a chance I missed something. Although I have a good feel for this area in general, I did not carefully check the paper's details, e.g., the math, experimental design, or novelty.

---

> ### Author Rebuttal · Authors · 2023-08-29
>
> We sincerely thank the reviewer for valuable comments. We use RR.n and Q.n to denote the response to “Reasons To Reject,” or ``Questions For The Authors” n. For the reviewers’ convenience, we retype the review comments and suggestions in bold followed by our responses. The response will be clarified in the revised paper.
>
> **RR.1: Figure 1 is hard to follow. There is no caption for Figure 1, really can not understand difference between two step 5 and two step 1.**
>
> RR.1 Response: Thanks for the valuable suggestion. We apologize for the confusion. We should have put Step 5 below Step 3. In Step 1, the devices receive the corresponding updated prompt parameters of specific layers from the server. In Step 5, the prompt parameters of specific layers are sent back to the server.
>
> Within traditional federated learning, e.g., FedAvg, each round consists of 5 steps. First, the server randomly selects a set of devices. Second, the server broadcasts the models to selected devices. Third, the selected devices perform local model updates. Fourth, the selected devices upload the updated local models. Fifth, the server performs the aggregation of received models. Step 1 in FedPepTAO is similar to the second step of FedAvg and Step 5 in FedPepTAO is similar to the fourth step of FedAvg. This will be clarified in the revised version.
>
> **RR.2 + Q: Table 2, to reach a target acc, reviewer wanna know how those acc are set as target? We know the time to achieve an acc may be vary for different factors. Reviewer believe use a epoch and communication overhead will be more reasonable. How many times did the authors did this experiment? How do authors get those number, the best runtime? Avg runtime?**
>
> RR.2 + Q Response: Thanks for the thoughtful comment. We select the highest target accuracy that at least 5 baseline approaches can achieve. We repeated the experiment with three random seeds (32, 42, 52) and calculated the average best accuracy, e.g., 22.38\%, 7.22\%, 24.07\%, 5.68\%, 13.84\%, 1.32\% higher compared to FedPrompt, P-tuning v2, ATTEMPT, LPT, MomD and MomS+AdamD, with RoBERTa-Large model on RTE dataset.
>
>
> We conducted experiments with the number of epochs required to achieve the target accuracy and the communication overhead to demonstrate the performance of FedPepTAO on the RoBERTA-Large model and 10 tasks. Below are the average epochs required to achieve the target accuracy.
>
> Table 1. The average number of epochs required to achieve the target accuracy (85% for QNLI, 92.5% for SST-2, 3% for CoLA, 77% for MRPC, 65% for RTE, 71% for BoolQ, 85% for MPQA, 88% for Subj, 78% for Trec, 91% for MR). "/" represents that training does not achieve the target accuracy.
> | Method        | Epochs      |
> |---------------|-------------|
> | Adapter       | 30.92       |
> | FedPrompt     | 24.50       |
> | P-tuning v2   | 5.99        |
> | Prompt Tuning | /           |
> | IDPG          | 40.15       |
> | ATTEMPT       | 39.97       |
> | LPT           | 8.31        |
> | MomD          | 13.98       |
> | MomS+AdamD    | 5.99        |
> | FedPepTAO     | 3.88        |
>
>
> From Table 1, we find that FedPepTAO requires the smallest amount of epochs to achieve the target accuracy (87.45\%, 84.16\%, 35.26\%, 90.33\%, 90.29\%, 53.31\%, 72.22\%, 35.21\% faster compared to Adapter, FedPrompt, P-tuning v2, IDPG, ATTEMPT, LPT, MomD, and MomS+AdamD respectively). Prompt Tuning failed to achieve the target accuracy since it only optimizes the first layer of soft prompts. FedPepTAO can also reduce the communication overhead from 40\% to 41.55\% (41.55\%, 40\%, 40\%, 40\% compared with Adapter, P-tuning v2, MomD, and MomS+AdamD, respectively) between devices and the server, as illustrated in Table 2. FedPrompt, Prompt-Tuning, IDPG, ATTEMPT, and LPT correspond to lower communication overhead (up to 13\%) since they only select one layer during tuning, which results in significantly inferior accuracy (from 17.02\% to 60.8\% lower) compared with FedPepTAO. This will be clarified in the revised version.
>
> Table 2. The communication overhead (s) between devices and the server with RoBERTA-Large model.
> | Method       | Time  |
> |--------------|-------|
> | Adapter      | 5.80  |
> | FedPrompt    | 3.09  |
> | P-tuning v2  | 5.65  |
> | Prompt Tuning| 3.00  |
> | IDPG         | 3.09  |
> | ATTEMPT      | 3.16  |
> | LPT          | 3.09  |
> | MomD         | 5.65  |
> | MomS+AdamD   | 5.65  |
> | FedPepTAO    | 3.39  |
>
> **RR.3: They only has one model, which sounds not sufficient.**
>
> RR.3 Response: Thanks for your helpful comment. To demonstrate the adaptability of FedPepTAO, we carried out extra experiments with three additional decoder-based models, i.e., GPT-2 model (774M) on MRPC, MR, SST-2 dataset, LLaMA 3B model on RTE, MRPC dataset, and LLaMA 7B model on MRPC dataset. As shown in Table 3 below, FedPepTAO significantly outperforms baseline methods in terms of the best accuracy on the decoder-based GPT-2 model (up to 32.3\%, 18.23\%, 43.32\%, 15.94\%, 4\%, 4.34\% higher compared to FedPrompt, P-tuning v2, ATTEMPT, LPT, MomD and MomS+AdamD, respectively). Furthermore, the efficiency of FedPepTAO is significantly higher than baseline approaches (up to 84.74\%, 86.89\%, and 15.27\% faster compared to LPT, MomD, and MomS+AdamD, respectively).
>
> Table 3. Accuracy and tuning time (s) to achieve target accuracy (75% for MRPC, 81% for MR, and 92.5% for SST-2) on GPT-2 model. "/" represents that training does not achieve the target accuracy.
> | Method          | MRPC Acc | MRPC Time | MR Acc | MR Time | SST-2 Acc | SST-2 Time |
> |-----------------|---------|-----------|--------|---------|-----------|------------|
> | FedPrompt       | 74.98   |     /     | 57.2   |    /    | 76.49     |     /      |
> | P-tuning v2     | 74.8    |     /     | 73     |    /    | 74.77     |     /      |
> | ATTEMPT         | 37.91   |     /     | 57.3   |    /    | 85.89     |     /      |
> | LPT             | 74.98   |     /     | 84.8   |  1455   | 77.06     |     /      |
> | MomD            | 77.23   |   503     | 85.7   |  1694   | 92.55     |    3638    |
> | MomS+AdamD      | 76.89   |   291     | 88.2   |   262   | 92.55     |    2488    |
> | FedPepTAO       | 81.23   |   273     | 89.5   |   222   | 93        |    2248    |
>
> When the model becomes larger, i.e., LLaMA 3B with 3 billion parameters, FedPepTAO still achieves the best accuracy (up to 4.6\%, 11.29\%, 5.28\%, 6.69\%, 6.2\%, 10.88\% higher compared to FedPrompt, P-tuning v2, ATTEMPT, LPT, MomD and MomS+AdamD, respectively) and better efficiency (up to 39.81\%, 43.04\%, 48.16\%, 38.86\%, 9.72\% faster compared to FedPrompt, ATTEMPT, LPT, MomD, and MomS+AdamD, respectively) as illustrated in Table 4.
>
> Table 4. Accuracy and tuning time (s) to achieve target accuracy (75% for RTE, 81% for MRPC) on LLaMA 3B model. "/" represents that training does not achieve the target accuracy.
> | Method        | RTE Acc | RTE Time | MRPC Acc | MRPC Time |
> |---------------|---------|----------|----------|----------|
> | FedPrompt     | 78.34   | 540      | 81.86    | 459      |
> | P-tuning v2   | 56.68   |   /      | 75.17    |   /      |
> | ATTEMPT       | 64.98   |   /      | 81.18    | 718      |
> | LPT           | 64.98   |   /      | 79.77    | 789      |
> | MomD          | 80.87   | 360      | 80.26    | 669      |
> | MomS+AdamD    | 80.87   | 360      | 75.58    |   /      |
> | FedPepTAO     | 83.39   | 325      | 86.46    | 409      |
>
>
> We further verify the performance of our method with another large model, i.e., LLaMA 7B with 7 billion parameters. As presented in Table 5, FedPepTAO outperforms baseline methods in terms of accuracy (up to 5.05\%, 26.71\%, 18.41\%, 18.41\%, 6.2\%, 10.88\% higher compared to FedPrompt, P-tuning v2, ATTEMPT, LPT, MomD and MomS+AdamD, respectively) and efficiency (up to 15.77\%, 75.14\%, 32.06\%, 67.04\% faster compared to ATTEMPT, LPT, MomD, and MomS+AdamD, respectively), which demonstrates the scalability of FedPepTAO.
>
> Table 5. Accuracy and tuning time (s) to achieve target accuracy (75% for MRPC) on LLaMA 7B model. "/" represents that training does not achieve the target accuracy.
> | Method       | MRPC Acc | MRPC Time |
> |--------------|----------|-----------|
> | FedPrompt    | 76.18    |    /      |
> | P-tuning v2  | 74.98    |    /      |
> | ATTEMPT      | 81.52    |   317     |
> | LPT          | 81.95    |  1074     |
> | MomD         | 76.2     |   393     |
> | MomS+AdamD   | 75.75    |   810     |
> | FedPepTAO    | 82.34    |   267     |
>
> This will be clarified in the revised version.

---

### Official Review · Reviewer_6pQ8 · 2023-08-13

**Soundness:** 3

**Excitement:**

4: Strong: This paper deepens the understanding of some phenomenon or lowers the barriers to an existing research direction.

**Paper Topic And Main Contributions:**

The authors present an efficient prompt tuning strategy for Federated Learning of Large Language Models. The idea is to select (based on a scoring function) a subset of layers (instead of full set of prompt parameters) and exchange the prompt parameters of these layers during the tuning process. In addition to this, the authors present a communication efficient strategy to update the gradients on both server and device.

**Questions For The Authors:**

A. Are all the 10 Tasks used in the evaluation, encoder only tasks?

B. Is there a metric to compare the communication efficiency of this technique? (Other than the number of parameters?)

C. How do you evaluate and compare the client drift problem?

**Reasons To Accept:**

The work presented has a strong and important motivation to improve the efficiency and reduce the communication and computation costs in FL of LLMs. It provides an initial framework to combine ideas from efficient fine tuning strategies (like prompt-tuning and prefix-tuning) with Federated Learning. The approach taken to achieve this is novel and architecture agnostic, so the ideas are scalable to FL of any model architectures.

**Reasons To Reject:**

Although the approach is generic enough, I did not see the results of applying this approach to decoder models in the paper. All the results seemed to use RoBERT-Large and would have liked to see other variations like encoder-decoder and decoder only models as well.

**Reproducibility:**

3: Could reproduce the results with some difficulty. The settings of parameters are underspecified or subjectively determined; the training/evaluation data are not widely available.

**Reviewer Confidence:**

3: Pretty sure, but there's a chance I missed something. Although I have a good feel for this area in general, I did not carefully check the paper's details, e.g., the math, experimental design, or novelty.

---

> ### Author Rebuttal · Authors · 2023-08-29
>
> We sincerely thank the reviewer for valuable comments. We use RR.n and Q.n to denote the response to “Reasons To Reject,” or ``Questions For The Authors” n. For the reviewers’ convenience, we retype the review comments and suggestions in bold followed by our responses. The response will be clarified in the revised paper.
>
> **RR.1: Although the approach is generic enough, I did not see the results of applying this approach to decoder models in the paper. All the results seemed to use RoBERT-Large and would have liked to see other variations like encoder-decoder and decoder only models as well.**
>
> RR.1 Response: Thanks for the valuable comment. To demonstrate the adaptability of FedPepTAO, we carried out extra experiments with three additional decoder-based models, i.e., GPT-2 model (774M) on MRPC, MR, SST-2 dataset, LLaMA 3B model on RTE, MRPC dataset, and LLaMA 7B model on MRPC dataset.
>
> As shown in Table 1 below, FedPepTAO significantly outperforms baseline methods in terms of the best accuracy on the decoder-based GPT-2 model (up to 32.3\%, 18.23\%, 43.32\%, 15.94\%, 4\%, 4.34\% higher compared to FedPrompt, P-tuning v2, ATTEMPT, LPT, MomD and MomS+AdamD, respectively). Furthermore, the efficiency of FedPepTAO is significantly higher than baseline approaches (up to 84.74\%, 86.89\%, and 15.27\% faster compared to LPT, MomD, and MomS+AdamD, respectively).
>
> Table 1. Accuracy and tuning time (s) to achieve target accuracy (75% for MRPC, 81% for MR, and 92.5% for SST-2) on GPT-2 model. "/" represents that training does not achieve the target accuracy.
> | Method          | MRPC Acc | MRPC Time | MR Acc | MR Time | SST-2 Acc | SST-2 Time |
> |-----------------|---------|-----------|--------|---------|-----------|------------|
> | FedPrompt       | 74.98   |     /     | 57.2   |    /    | 76.49     |     /      |
> | P-tuning v2     | 74.8    |     /     | 73     |    /    | 74.77     |     /      |
> | ATTEMPT         | 37.91   |     /     | 57.3   |    /    | 85.89     |     /      |
> | LPT             | 74.98   |     /     | 84.8   |  1455   | 77.06     |     /      |
> | MomD            | 77.23   |   503     | 85.7   |  1694   | 92.55     |    3638    |
> | MomS+AdamD      | 76.89   |   291     | 88.2   |   262   | 92.55     |    2488    |
> | FedPepTAO       | 81.23   |   273     | 89.5   |   222   | 93        |    2248    |
>
> When the model becomes larger, i.e., LLaMA 3B with 3 billion parameters, FedPepTAO still achieves the best accuracy (up to 4.6\%, 11.29\%, 5.28\%, 6.69\%, 6.2\%, 10.88\% higher compared to FedPrompt, P-tuning v2, ATTEMPT, LPT, MomD and MomS+AdamD, respectively) and better efficiency (up to 39.81\%, 43.04\%, 48.16\%, 38.86\%, 9.72\% faster compared to FedPrompt, ATTEMPT, LPT, MomD, and MomS+AdamD, respectively) as illustrated in Table 2.
>
> Table 2. Accuracy and tuning time (s) to achieve target accuracy (75% for RTE, 81% for MRPC) on LLaMA 3B model. "/" represents that training does not achieve the target accuracy.
> | Method        | RTE Acc | RTE Time | MRPC Acc | MRPC Time |
> |---------------|---------|----------|----------|----------|
> | FedPrompt     | 78.34   | 540      | 81.86    | 459      |
> | P-tuning v2   | 56.68   |   /      | 75.17    |   /      |
> | ATTEMPT       | 64.98   |   /      | 81.18    | 718      |
> | LPT           | 64.98   |   /      | 79.77    | 789      |
> | MomD          | 80.87   | 360      | 80.26    | 669      |
> | MomS+AdamD    | 80.87   | 360      | 75.58    |   /      |
> | FedPepTAO     | 83.39   | 325      | 86.46    | 409      |
>
>
> We further verify the performance of our method with another large model, i.e., LLaMA 7B with 7 billion parameters. As presented in Table 3, FedPepTAO outperforms baseline methods in terms of accuracy (up to 5.05\%, 26.71\%, 18.41\%, 18.41\%, 6.2\%, 10.88\% higher compared to FedPrompt, P-tuning v2, ATTEMPT, LPT, MomD and MomS+AdamD, respectively) and efficiency (up to 15.77\%, 75.14\%, 32.06\%, 67.04\% faster compared to ATTEMPT, LPT, MomD, and MomS+AdamD, respectively), which demonstrates the scalability of FedPepTAO.
>
> Table 3. Accuracy and tuning time (s) to achieve target accuracy (75% for MRPC) on LLaMA 7B model. "/" represents that training does not achieve the target accuracy.
> | Method       | MRPC Acc | MRPC Time |
> |--------------|----------|-----------|
> | FedPrompt    | 76.18    |    /      |
> | P-tuning v2  | 74.98    |    /      |
> | ATTEMPT      | 81.52    |   317     |
> | LPT          | 81.95    |  1074     |
> | MomD         | 76.2     |   393     |
> | MomS+AdamD   | 75.75    |   810     |
> | FedPepTAO    | 82.34    |   267     |
>
> This will be clarified in the revised version.
>
> **Q.A. Are all the 10 Tasks used in the evaluation, encoder only tasks?**
>
> Q.A. Response: Thanks for raising this straightforward question. In our experiment, we exploited the 10 tasks with an encoder-only model. However, the 10 tasks are not encoder-only tasks and can be performed with decoder models. We carried out extra experiments with three additional decoder models, i.e., GPT-2 model (774M) on MRPC, MR, SST-2 dataset, LLaMA 3B model on RTE, MRPC dataset, and LLaMA 7B model on MRPC dataset.
>
> As shown in Table 1 below, FedPepTAO significantly outperforms baseline methods in terms of the best accuracy on the decoder-based GPT-2 model (up to 32.3\%, 18.23\%, 43.32\%, 15.94\%, 4\%, 4.34\% higher compared to FedPrompt, P-tuning v2, ATTEMPT, LPT, MomD and MomS+AdamD, respectively). Furthermore, the efficiency of FedPepTAO is significantly higher than baseline approaches (up to 84.74\%, 86.89\%, and 15.27\% faster compared to LPT, MomD, and MomS+AdamD, respectively).
>
> Table 1. Accuracy and tuning time (s) to achieve target accuracy (75% for MRPC, 81% for MR, and 92.5% for SST-2) on GPT-2 model. "/" represents that training does not achieve the target accuracy.
> | Method          | MRPC Acc | MRPC Time | MR Acc | MR Time | SST-2 Acc | SST-2 Time |
> |-----------------|---------|-----------|--------|---------|-----------|------------|
> | FedPrompt       | 74.98   |     /     | 57.2   |    /    | 76.49     |     /      |
> | P-tuning v2     | 74.8    |     /     | 73     |    /    | 74.77     |     /      |
> | ATTEMPT         | 37.91   |     /     | 57.3   |    /    | 85.89     |     /      |
> | LPT             | 74.98   |     /     | 84.8   |  1455   | 77.06     |     /      |
> | MomD            | 77.23   |   503     | 85.7   |  1694   | 92.55     |    3638    |
> | MomS+AdamD      | 76.89   |   291     | 88.2   |   262   | 92.55     |    2488    |
> | FedPepTAO       | 81.23   |   273     | 89.5   |   222   | 93        |    2248    |
>
> When the model becomes larger, i.e., LLaMA 3B with 3 billion parameters, FedPepTAO still achieves the best accuracy (up to 4.6\%, 11.29\%, 5.28\%, 6.69\%, 6.2\%, 10.88\% higher compared to FedPrompt, P-tuning v2, ATTEMPT, LPT, MomD and MomS+AdamD, respectively) and better efficiency (up to 39.81\%, 43.04\%, 48.16\%, 38.86\%, 9.72\% faster compared to FedPrompt, ATTEMPT, LPT, MomD, and MomS+AdamD, respectively) as illustrated in Table 2.
>
> Table 2. Accuracy and tuning time (s) to achieve target accuracy (75% for RTE, 81% for MRPC) on LLaMA 3B model. "/" represents that training does not achieve the target accuracy.
> | Method        | RTE Acc | RTE Time | MRPC Acc | MRPC Time |
> |---------------|---------|----------|----------|----------|
> | FedPrompt     | 78.34   | 540      | 81.86    | 459      |
> | P-tuning v2   | 56.68   |   /      | 75.17    |   /      |
> | ATTEMPT       | 64.98   |   /      | 81.18    | 718      |
> | LPT           | 64.98   |   /      | 79.77    | 789      |
> | MomD          | 80.87   | 360      | 80.26    | 669      |
> | MomS+AdamD    | 80.87   | 360      | 75.58    |   /      |
> | FedPepTAO     | 83.39   | 325      | 86.46    | 409      |
>
>
> We further verify the performance of our method with another large model, i.e., LLaMA 7B with 7 billion parameters. As presented in Table 3, FedPepTAO outperforms baseline methods in terms of accuracy (up to 5.05\%, 26.71\%, 18.41\%, 18.41\%, 6.2\%, 10.88\% higher compared to FedPrompt, P-tuning v2, ATTEMPT, LPT, MomD and MomS+AdamD, respectively) and efficiency (up to 15.77\%, 75.14\%, 32.06\%, 67.04\% faster compared to ATTEMPT, LPT, MomD, and MomS+AdamD, respectively), which demonstrates the scalability of FedPepTAO.
>
> Table 3. Accuracy and tuning time (s) to achieve target accuracy (75% for MRPC) on LLaMA 7B model. "/" represents that training does not achieve the target accuracy.
> | Method       | MRPC Acc | MRPC Time |
> |--------------|----------|-----------|
> | FedPrompt    | 76.18    |    /      |
> | P-tuning v2  | 74.98    |    /      |
> | ATTEMPT      | 81.52    |   317     |
> | LPT          | 81.95    |  1074     |
> | MomD         | 76.2     |   393     |
> | MomS+AdamD   | 75.75    |   810     |
> | FedPepTAO    | 82.34    |   267     |
>
> This will be clarified in the revised version.
>
> **Q.B. Is there a metric to compare the communication efficiency of this technique? (Other than the number of parameters?)**
>
> Q.B. Response: Thanks for the valuable comment. We utilized the tuning time to achieve target accuracy as a metric. As shown in Table 2 of the paper, FedPepTAO can outperform baseline approaches from 80.48\% to 99\% (95.91\%, 95.17\%, 92.76\%, 99\%, 97.28\%, 97.59\%, 85.8\%, 94.23\%, 80.48\% compared with Adapter, FedPrompt, P-tuning v2, Ptompt Tuning, IDPG, ATTEMPT, LPT, MomD, and MomS+AdamD, respectively). In addition, we measured the time to transfer the parameters between devices and the server:
>
> Table 4. The communication overhead (s) between devices and the server with RoBERTA-Large model.
> | Method       | Time |
> |--------------|-------|
> | Adapter      | 5.80  |
> | FedPrompt    | 3.09  |
> | P-tuning v2  | 5.65  |
> | Prompt Tuning| 3.00  |
> | IDPG         | 3.09  |
> | ATTEMPT      | 3.16  |
> | LPT          | 3.09  |
> | MomD         | 5.65  |
> | MomS+AdamD   | 5.65  |
> | FedPepTAO    | 3.39  |
>
> From Table 4, we find that FedPepTAO can significantly reduce the communication time from 40\% to 41.55\% (41.55\%, 40\%, 40\%, 40\% compared with Adapter, P-tuning v2, MomD, and MomS+AdamD, respectively) between devices and the server thanks to our parameter-efficient prompt tuning method. The communication overhead of FedPrompt, Prompt Tuning, IDPG, ATTEMPT, and LPT are slightly lower (up to 13\%), but they optimize only one prompt layer among all the layers, resulting in a significantly lower accuracy (from 17.02\% to 60.8\% lower) compared with FedPepTAO. This will be clarified in the revised version.
>
> **Q.C. How do you evaluate and compare the client drift problem?**
>
> Q.C. Response: Thank you for the insightful question. As a distributed optimization method, FL has a key challenge: the heterogeneity (non-iid-ness) in the data present on the diverse clients. Previous research [1] has shown that such heterogeneity introduces a drift in the updates of each client, i.e., client drift, resulting in inferior and unstable convergence. Hence, we evaluate the client drift problem based on the final accuracy with diverse approaches. As shown in Table 1 of the paper, FedPepTAO can outperform baseline approaches from 13.39\% to 60.8\% (25.39\%, 23.83\%, 14.53\%, 60.8\%, 51.76\%, 51.72\%, 17.02\%, 54.71\%, 13.39\% compared with Adapter, FedPrompt, P-tuning v2, Prompt Tuning, IDPG, ATTEMPT, LPT, MomD, and MomS+AdamD, respectively). In addition, we compare the single module of our communication-efficient adaptive optimization with other adaptive optimization baseline approaches without states for each device, which demonstrates significant advantages (up to 29.24\% compared with FedAvg, 8.3\% compared with MomD, 29.24\% compared with MomS, 29.24\% compared with MomS+Con, 2.16\% compared with AdamD, 5.05\% compared with MomS+AdamD higher in terms of accuracy) of our method, as shown in Figure 4 of the paper. Furthermore, we carried out extra experimentation to show that the value of the state ($c_i^r$) on each device is 35.91\% (on average) compared with that of the momentum ($m_i^r$), which reveals that the state for each device is effective while addressing the client drift problem. This will be clarified in the revised version.
>
> [1] Sai Praneeth Karimireddy, Satyen Kale, Mehryar Mohri, Sashank Reddi, Sebastian Stich, and Ananda Theertha Suresh. 2020b. Scaffold: Stochastic controlled averaging for federated learning. Int. Conf. on Machine Learning (ICML), pages 5132–5143. PMLR.

---

### Meta-Review · Area_Chair_DPgA · 2023-09-15

**Recommendation:** 4

**Metareview:**

This paper proposes techniques for improving the efficiency of prompt tuning in a federated learning setting. The core of the method is around selecting a subset of most impactful layers to update, and efficiently communicating the gradients. Reviewers found the proposed methods solid and novel, the experiments to be mostly convincing, and the problem and approach to be of high interest to the community. Most of the concerns were around the scope of experiments, which were mostly addressed by the new results provided during the rebuttal and will be added to the paper. I will only add that the evaluation can be cleaner if some confidence intervals were reported, and also that I was surprised not to see a comparison to LoRA for a parameter efficient training method.

---

### Decision · Program_Chairs · 2023-10-07

**Decision:**

Accept-Main

**Comment:**

This paper proposes techniques for improving the efficiency of prompt tuning in a federated learning setting. The core of the method is around selecting a subset of most impactful layers to update, and efficiently communicating the gradients. Reviewers found the proposed methods solid and novel, the experiments to be mostly convincing, and the problem and approach to be of high interest to the community. Most of the concerns were around the scope of experiments, which were mostly addressed by the new results provided during the rebuttal and will be added to the paper. I will only add that the evaluation can be cleaner if some confidence intervals were reported, and also that I was surprised not to see a comparison to LoRA for a parameter efficient training method.